# Structure, Growth and Histology of Gnathal Elements in *Dunkleosteus* (Arthrodira, Placodermi), with a Description of a New Species from the Famennian (Upper Devonian) of the Tver Region (North-Western Russia)

Oleg A. Lebedev [1], Russell K. Engelman [2], Pavel P. Skutschas [3], Zerina Johanson [4,*], Moya M. Smith [4,5], Veniamin V. Kolchanov [3], Kate Trinajstic [6] and Valeriy V. Linkevich [7]

[1] A. A. Borissiak Palaeontological Institute of the Russian Academy of Sciences, Moscow 117997, Russia; elops12@yandex.ru

[2] Department of Biology, Case Western Reserve University, Cleveland, OH 44106, USA; neovenatoridae@gmail.com

[3] Department of Vertebrate Zoology, Saint Petersburg State University, Saint Petersburg 199034, Russia; p.skutschas@spbu.ru (P.P.S.)

[4] Natural History Museum, London SW7 5BD, UK

[5] Centre for Craniofacial Regenerative Biology (CCRB), Faculty of Dentistry, Oral and Craniofacial Sciences, King's College, London SE1 1UL, UK

[6] School of Molecular and Life Sciences, Curtin University, Perth 6845, Australia; k.trinajstic@curtin.edu.au

[7] E. E. Shimkevich Andreapol Historical and Natural History Museum, Andreapol 172800, Russia; linkevichvalerijj@rambler.ru

[*] Correspondence: z.johanson@nhm.ac.uk

**Abstract:** A new species of *Dunkleosteus*, *D. tuderensis* sp. nov., is named based on an infragnathal from the Famennian of the Tver Region, Russia. CT scanning of the holotype revealed two high-density bony constituents comparable in position and interrelations to components described for coccosteomorph arthrodires, supported by the presence of at least two clusters of large vascular canals marking separate arterial supplies. Coccosteomorph and dunkleosteid pachyosteomorphs exhibit similar growth patterns including labio-basal depositions of vascularized bone in the infragnathals and basally in the supragnathals. In contrast to coccosteomorphs, dunkleosteid reinforcement of the occlusal margins occurred via the formation of dense osteonal bone, in parallel with resorption forming extensive lingual fossae. Active bone remodeling proceeded without a complete reworking of the primary osteonal bone structure and the original arrangement of vascular canals. Due to inconsistent anatomical terminology in gnathal elements of dunkleosteid arthrodires, a revised terminology is suggested and new terms are introduced.

**Keywords:** arthrodires; morphology; histology; gnathals; odontoid; oral component; para-articular component; Bilovo

## 1. Introduction

Dunkleosteid arthrodires, some of the largest predators in the Devonian period [1], have a peculiar oral structure in which the dermal jaw elements (gnathal plates) are sharpened to process food [2–4]. However, despite their imposing nature, these structures have been only superficially studied, as most morphological studies on this group have focused upon the structure of the skull roof and the cheek complex (for example, [5]). Dunkleosteid gnathal elements are usually described only in gross features (for example, [6–12] and others), whereas recent functional studies of arthrodire jaw mechanisms are mostly based on computerized models [13–17] and do not focus on specific anatomic details.

Histological studies of the jaw elements of dunkleosteid arthrodires have been limited and consider few aspects of their complicated nature [10,18,19] or are focused on special

problems of the presence of dentine and bone remodeling [20]. However, the study of dunkleosteid jawbone histology is important in terms of these animals' evolutionary biology and functional morphology. Adult arthrodire placoderms are the only vertebrates in which physical contact with the surrounding environment during feeding and food processing is carried out directly by their jaw bones, rather than by teeth or keratinized beaks. However, superficial bone layers exposed in the mouth show little signs of stress and damage, and evidence of infection is lacking. This raises a large number of questions as to how the gnathal elements of dunkleosteid arthrodires worked. How did these animals deal with potential infection when the vascular canals of the gnathals became exposed by occlusal wear or damage? How did the bone not bleed or feel pain? How did the gnathal elements function with a disrupted periosteum, which is necessary for bones to grow or heal? These questions cannot be answered without a better understanding of the structure and function of arthrodire gnathals. The distribution patterns of various bone tissues in pachyosteomorph arthrodires have only been studied in very general terms, although these data are of primary importance in studies of skeletal ontogeny.

New data from reasonably well-preserved dunkleosteid gnathals from Russia provide an opportunity for new insights into this field of arthrodire research. Previous descriptions of dunkleosteid remains from the East European platform are available, based upon scarce and fragmentary specimens. These finds are limited geographically to the central part of Russia, historically termed in the geological literature as the 'Central Devonian Field' (CDF). O. Obrucheva [21] established a new species, "*Dinichthys*" *machlaevi*, from the Upper Famennian of the Orel Region based on an isolated posterior supragnathal plate. This taxon had been conditionally referred to *Dunkleosteus* by Denison [22]. Since that time, new dunkleosteid specimenshave been collected from this locality and will be described elsewhere. A nuchal plate, designated by Obrucheva [23] as *Dinichthys* sp. 2, from the lower Famennian of the same region, has been suggested to pertain to the genus *Eastmanosteus* [24,25]. Lebedev et al. [26] described an incompletely preserved antero-lateral plate of the thoracic armor from the Khovanshchinian Regional Stage (lower uppermost Famennian) of the Lipetsk Region and identified it as ?Dunkleosteidae gen. et sp. indet. Moloshnikov [24] presented an incomplete anterior ventro-lateral plate belonging to Dunkleosteidae gen. et sp. indet. from the lower Famennian of the Gornostayevka quarry (Livny Lime factory) close to the town of Livny (Orel Region). Apart from these descriptions, several papers have mentioned other 'pachyosteomorph' or 'dunkleosteid' remains from Russia (for example, [27–29]). Thus, new dunkleosteid specimens from Russia provide a wealth of new systematic and morphological information, and are important for paleozoogeographical research, significantly increasing the known area inhabited by these fishes and providing new data on interprovincial faunal connections.

This new specimen has been collected from one of the few Famennian vertebrate localities yielding abundant, well-preserved material in the eastern part of northwest Russia, traditionally named the 'Main Devonian Field' (MDF). Here, several sections occur along the banks of the Maliy Tuder River in the north-west of the Tver Region (Figure 1A). The list of vertebrate taxa from these localities was first published by Obruchev [30] and since then has been only slightly modified in the stratigraphical literature (for example, [31]) and expanded in further paleontological studies.

Since 2011, the exposures along the banks of the Lovat' River tributaries in the Andreapol and Toropets Districts of the Tver Region have been studied by members of the E.E. Shimkevich Historical and Natural History Museum of Andreapol. Their research is aimed at collecting Upper Devonian fossils. More recent collecting from the Bilovo locality resulted in a number of recent papers describing new taxa, and revising previously collected material of brachiopods [32], cephalopods [33], antiarchs [34–37], dipnoans [38] and chondrichthyans [39].

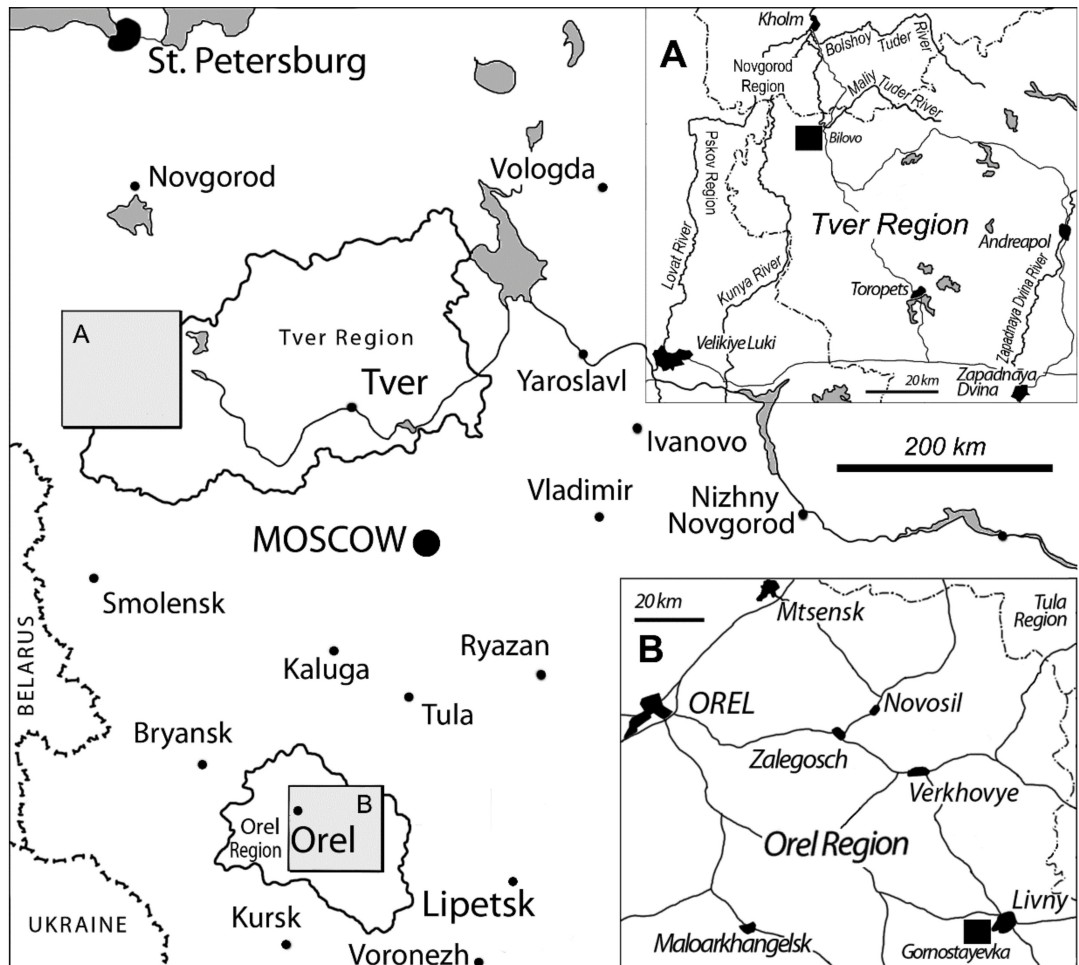

**Figure 1.** Geographical position of the Bilovo locality (**A**), type locality for *Dunkleosteus tuderensis* Lebedev, sp. nov., and Gornostayevka locality (**B**), which yielded materials used for histological studies.

　　In 2015, the vertebrate collection of the Museum was enriched by Daniil V. Linkevich, who discovered one of the first arthrodire specimens from the MDF (the first being from the Tērvete Formation of Latvia; [40]), a dunkleosteid, from Famennian outcrops in the Tver Region, Russia, adding to our knowledge on the composition and structure of Famennian vertebrate assemblages in this region.

　　The present paper describes the general morphology and microanatomy of this specimen, as well as establishing its systematic position. High-quality preservation made it possible to perform micro-CT studies of this material, providing data on its internal structure. Unfortunately, the uniqueness of this specimen meant it could not be sacrificed for the preparation of thin sections. Discussion of the taxonomic status of the new specimen involved re-examination of the morphology of some specimens referred to as *Dunkleosteus terrelli* from the collections of the Cleveland Museum of Natural History (Cleveland, OH, USA).

　　In order to obtain new histological information on the dunkleosteid gnathal elements, we used additional materials, assigned here to Dunkleosteidae gen. et sp. indet., from the Lower Famennian Gornostayevka locality in the Orel Region (CDF), found in the deposits of a presumably similar age based on the similarity of the fish assemblages (OL, pers. obs., Figure 1B).

## 2. Materials and Methods

　　Materials described in the present paper include an incomplete right infragnathal (specimen KMA 4155) lacking the adductor lamina, as well as several other infragnathals

and the anterior and posterior supragnathals from the Gornostayevka locality in the Orel Region of Russia. The latter were used for histological analysis and are recorded in the collection PIN 2657 (PIN 2657/371, section PIN 2657/371a, b; PIN 2657/376, section PIN 2657/376a; PIN 2657/385, sections PIN 2657/385a, b; PIN 2657/389, section PIN 2657/389a, b; PIN 2657/390, sections PIN 2657/390a, b).

For comparative purposes, additional observations of late Devonian arthrodires were drawn from specimens housed at the Cleveland Museum of Natural History (CMNH) and American Museum of Natural History (AMNH). Observations of the CMNH material were made both in person and using 3D surface scans published by the CMNH on Morphosource (www.morphosource.org). Material of other arthrodires from Cleveland Shale (*Gorgonichthys*, *Heintzichthys*, *Bungartius*) was also examined directly from the collections of the CMNH. Observations on material from the Gogo Formation were drawn from material housed at the Western Australian Museum (WAM). Other anatomical observations on arthrodires were made using the previously published literature.

Usually, specimen preservation is only described briefly. To enrich our morphological analysis, we also describe the microanatomy of the superficial bone structure of KMA 4155, constructing a map of the damaged areas and distinguishing those areas showing greater wear from those areas only slightly worn or intact (Supplementary Figure S1). The specimen is slightly worn superficially; there are several fractures, and minor portions of the occlusal margin are missing. The occlusal margin and the originally acute margins of the broken surface posteriorly are somewhat rounded. The ventral bone margin, including the antero-ventral flange, has been destroyed by abrasion. Apart from this, an irregular area mostly adjoining the occlusal facet (extending ventrally from the occlusal margin) has been damaged by probable chemical weathering or bioerosion. The lingual face of the specimen demonstrates only some abrasion ventrally, as well as along the occlusal margin.

The preservation of additional material from the Gornostayevka quarry (collection PIN 2657) is highly variable and strongly dependent on the lithological association of the fossils. As the arthrodire skeletal elements had been found within a wide interval of the section, including loosely cemented sands or sands cemented with ferrous minerals, clays and limestones [29], preservation of bone tissue is variably affected by diverse weathering agents. Whereas the inner structure mostly remained unaltered and so is appropriate for histological studies, the extent of erosion of the superficial bone layers present in some specimens needs to be taken into consideration. The specimens with the best internal and external preservation are those found at the clay/limestone contact, whereas specimens recovered from sands are often permeated by ferrous minerals, like hematite, goethite and hydrogoethite, and/or the superficial bone layers are extensively altered.

Micro-CT tomography was carried out on the NEOSCAN 80 in the A.A. Borissiak Paleontological Institute of the Russian Academy of Science (PIN), Moscow, Russia, using software Version 2.2.4. The scan parameters for KMA 4155 were as follows: source voltage, 110 kV; source current, 146 μA; camera exposure, 620 ms; filter, Cu, 1.0 mm; image pixel size, 37.375000 μm; and rotation step, 0.200°, in three connected scans. The characteristics for anterior supragnathal PIN 2657/388 scanning were as follows: source voltage, 101 kV; source current, 159 μA; camera exposure, 380 ms; filter, Cu, 0.5 mm; image pixel size, 23.920268 μm; and rotation step, 0.200°. The characteristics for anterior supragnathal PIN 2657/378 scanning were as follows: source voltage, 110 kV; source current, 146 μA; camera exposure, 620 ms; filter, Cu, 1.0 mm; image pixel size, 35.506504 μm; and rotation step, 0.200°. The characteristics for posterior supragnathal PIN 2657/386 scanning were as follows: source voltage, 101 kV; source current, 159 μA; camera exposure, 380 ms; filter, Cu, 0.5 mm; image pixel size, 26.731746 μm; and rotation step, 0.200°. Reconstructions were made by NeoScan software, version 2.3.2.

Preparation of specimen KMA 4155 was carried out manually using a mounted needle to remove the matrix. Most of the specimens from the Gornostayevka locality were manually prepared, although some were first etched using a solution of 10% acetic acid.

Thin sections from the fragmentary skeletal elements (anterior supragnathal PIN 2657/389, posterior supragnathal PIN 2657/390 and fragments of three infragnathals PIN 2657/371, 2657/376 and 2657/385) from the Gornostayevka quarry (Orel Region) were made using the standard grinding technique. The sections were examined under normal and polarized light using a Leica 4500 optical microscope in the Resource Centre "X-ray Diffraction Methods of Research" of the St. Petersburg State University Science Park (St. Petersburg, Russia). Histological terminology follows that suggested by Francillon-Vieillot et al. [41] and Johanson and Smith [20].

*Institutional Abbreviations*

AMNH, American Museum of Natural History, NY, USA; CCNHM, Mace Brown Museum of Natural History, Charleston, VA, USA; CMNH, Cleveland Museum of Natural History, Cleveland, OH, USA; KMA, E.E. Shimkevich Andreapol Historical and Natural History Museum (Andreapol, Tver Region, Russia); MMMN, Manitoba Museum of Man and Nature, Winnipeg, Canada; NHMUK PV P, The Natural History Museum, London, UK; PIN RAS, A.A. Borissiak Palaeontological Institute of the Russian Academy of Sciences, Moscow, Russia; WAM, Western Australian Museum, Perth, Australia.

## 3. Geographical and Geological Setting

The Bilovo group of localities includes several outcrops from both banks at the bend of the Maliy Tuder River (Figure 1A). These outcrops expose sandy, carbonaceous and clayey-sandy rocks of the Tuder, Bilovo and Lnyanka Formations, corresponding to the Eletsian, Lebedyanian and Optukhovian Regional Stages of the regional stratigraphic chart for the East European platform [31].

KMA/4155 was recovered from an outcrop on the left bank of Maliy Tuder River, about 200 m north–northwest from the currently abandoned Bilovo village (Toropets District, Tver Region, Russia; 56.90702266099799 N, 31.21591178227463 E). Only general lithological information about this site is available: the limestone member at the base is separated from overlaying variegated clays by a bed of light greyish blue loose sands, which yielded the specimen.

The geologic age of the Bilovo Formation is a matter of debate. It was originally considered Lebedyanian in age based on the presence of incompletely preserved spiriferid brachiopod *Cyrtospirifer* cf. *lebedianicus* [31]. The species was described from the deposits of the Lebedyanian Regional Stage (RS) of the CDF. In this territory, the Lebedyanian RS correlates with the upper *marginifera–trachytera* interval of the Standard conodont zonation (SCZ) [42]. However, Davydov and Linkevich [32] regarded identification of these brachiopods as erroneous and established a new species, *Cyrtospirifer biloviensis*, for this material. These authors noted that the new species is only rarely present in the territory of the CDF and thus in the case of the Bilovo Formation brachiopods, these cannot be reliably used to identify its age as Lebedyanian. Nevertheless, this currently seems to be the most reliable interpretation.

Shchedukhin [33] identified an assemblage of nautiloid cephalopods from the Bilovo Formation. He noted that the genus *Onyxites* had previously been recorded only from the lower Famennian (Zadonskian and Eletsian Regional Stages, upper *triangularis*—lower *marginifera* SCZ interval) of the CDF while the other species from his list of cephalopods are distributed within wider geochronological limits.

Lebedev et al. [39] described a fragment of a chondrichthyan fin spine assigned to *Ctenacanthus* aff. *venustus*, from the carbonaceous unit of these sections. In the East European platform, this species is otherwise known from the Eletsian Regional Stage (lower Famennian) of the CDF. Thus, an older age for the Bilovo Formation cannot be excluded, and this opinion matches that of Shchedukhin [33].

Thus, taking into account the present state of knowledge, the age of the Bilovo sections may be estimated within the wide range of the Eletsian–Lebedyanian RS corresponding

to *rhomboidea*—lower *trachytera* SCZ (Lower-Middle Famennian) (correlation of regional stages to SCZ after [42]).

Invertebrate remains found in the Bilovo sections include bivalve molluscs *Kochia tuderi*; *Schizodus tuderi* and *Posidonomya gibbosa*; gastropods "*Pleurotomaria*" *baschkirica*, *Naticopsis* sp., "*Murchisonia*" sp. and "*Cyrtolites*" sp.; nautiloid cephalopods: *Deiloceras evanidum*, ?*Archiacoceras inversum*, *Onyxites* sp. and Discosoridae indet.; inarticulate brachiopod valves of the lingulid type, rhynchonellid brachiopods *Ripidiorhynchus* ex gr. *livonicus*, spiriferid brachiopods *Cyrtospirifer biloviensis*; phyllocariid crustaceans *Echinocaris tudrensis*; ostracods *Cryptophyllus socialis*; unidentifiable microconchs earlier assigned to sedentary worms "*Serpula vipera*" and "*Spirorbis omphaloides*"; charophyte gyragonite endocasts; ferruginized worm trace fossils; and conodonts (under study now). This list is composed after Hecker and Filippova [30], Hecker [43], Sammet [44], Davydov and Linkevich [32] and Shchedukhin [33] with the addition of information obtained by O. Lebedev from studying microremains. The species of bivalves, gastropods and rhynchonellid brachiopods await revision.

The list of vertebrate taxa from the Bilovo Formation, including earlier published data [36,38] and newly obtained information include scales of acanthodians "*Devononchus*" *tenuispinus*? and *Acanthodes* sp.; scales of chondrichthyans (Hybodontida?); cranial and postcranial bones of antiarchs *Livnolepis heckeri* and *Bothriolepis* sp.; fragments of cranial and trunk bones of dunkleosteid pachyosteomorph arthrodires; postcranial elements of Ptyctodontidae; tooth plates, cranial and postcranial bones of dipnoans *Anchidipterus dariae* and a yet to be named dipterid, various cranial bones and scales of the osteolepiform *Megapomus heckeri*; scales of Tristichopteridae?, the onychodontiform *Strunius* sp.; and scales and teeth of indeterminate Actinopterygii.

Taking into account the lithological and faunal data, we regard the environmental conditions under which sedimentation occurred as shallow-water marine, influenced by some input of fresh waters, either periodical or constant.

## 4. Results

### 4.1. General Morphology and Terminology

The gnathal bones of dunkleosteid arthrodires have been studied for more than 150 years, yet morphological terminology for these skeletal elements remains inconsistent. Nomenclatorial problems with respect to these skeletal elements have never been addressed. For this reason, we propose to standardize the nomenclature, at least for the infragnathal bone, for this group of placoderms. Among dunkleosteid arthrodires, the best-known species is *Dunkleosteus terrelli*, which is used here as a model for this morphological study (Figure 2). Suggested changes especially concern those morphological structures which had been named previously according to their function or shape, for example, "occlusal" and "non-occlusal" [45], "biting" and "spatulate region" [12], or even "biting" and "non-biting" region (for example, [11,46] and others), "occlusal margins" [10], etc. Perhaps most confusing is the use of the term "blade" for the posterior, non-oral region of the infragnathal [10,11,47,48], which refers to the entire infragnathal except the part containing the blade-like oral margin (which other authors refer to as the "blade"; [22]).

The bony infragnathal of arthrodires consists of two major parts: an oral division and an adductor division (Figure 2A). The most prominent features on the oral margin are vertical structures commonly referred to as "tusk", "fang", "peak" or "cusps". At the posterior end of the occlusal margin, there is a triangular elevation sometimes bearing a row of teeth on its posterior slope. The two opposing infragnathal bones of the lower jaw did not fuse or directly contact the symphysis, with the rami being interconnected by the mentomandibular element or possibly the medial cartilaginous basimandibular element ventrally [11], although the presence of the latter has never been proven. In most arthrodires, the two halves of the infragnathal indirectly articulated through interlocking adsymphyseal (or symphyseal) teeth [49], but in dunkleosteids these teeth are absent. The

adductor division with attached Meckel's cartilages served as a third-class lever, where attached muscles effected the adduction of the lower jaw.

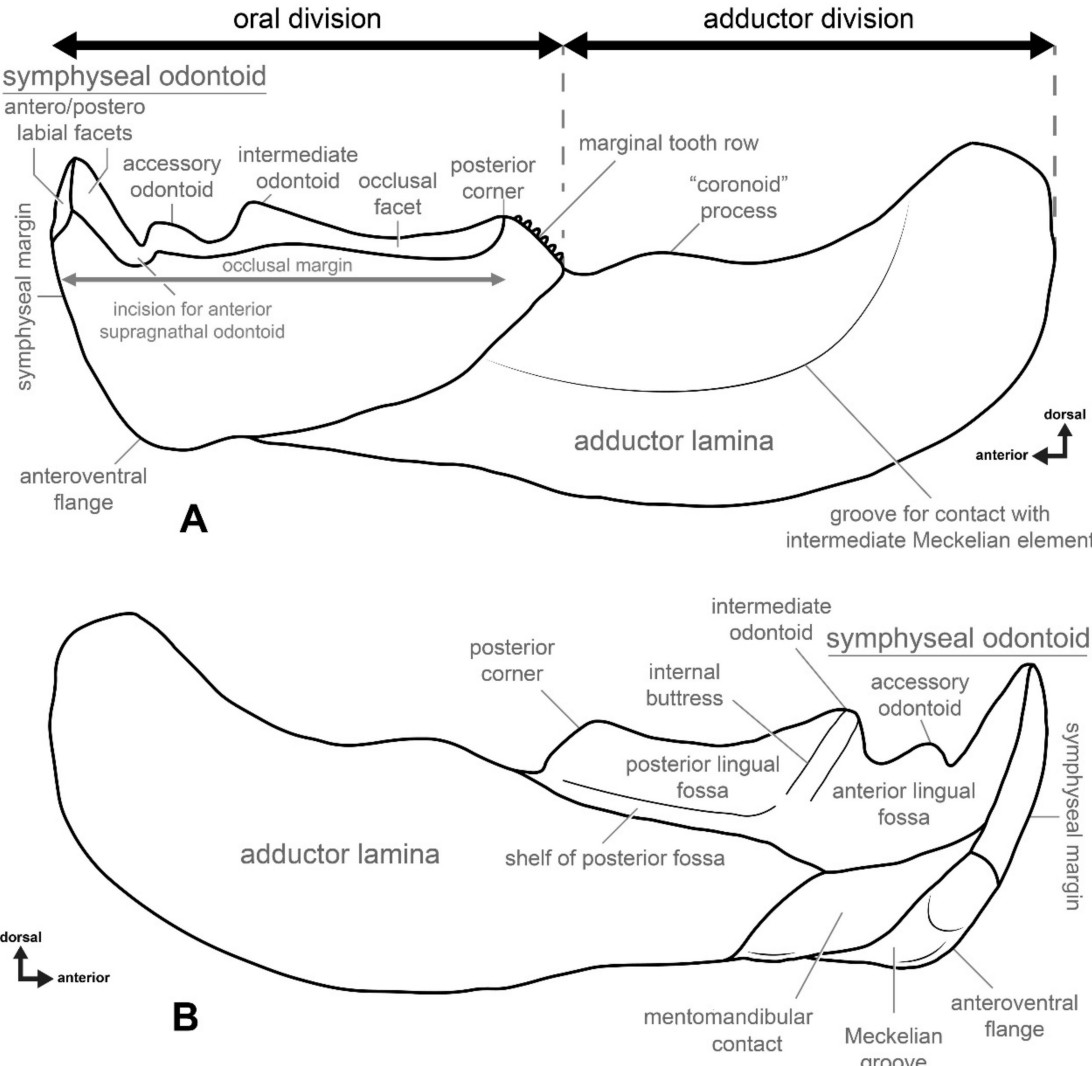

**Figure 2.** Morphology of a dunkleosteid infragnathal (left side) based on *Dunkleosteus terrelli* (modeled after CMNH 8808) and suggested terminology. (**A**) in labial and (**B**) in lingual view.

The terms used for structures on the occlusal margin also need revision. The terms "tusk", "fang", "peak" or "cusps", previously used, have a very narrow odontological meaning and referring to the structures in arthrodires as such creates confusion in respect to the homology and morphology, especially as these have a significantly different histological composition (see Discussion, below). To overcome issues regarding the non-homology of oral structures, we suggest the term "odontoid" be used to refer to the bony tooth-like projections seen in the jaws of arthrodires. Odontoid is a term used to refer to bony projections found on the lower jaws of some extant toothless amphibians, such as ceratophyrids, myobatrachids, hylids, ranids and leptodactylids [50], and as with the structures of arthrodires, are cusp- or fang-like outgrowths of the jaw bone used in prey capture and/or defense. This term makes it possible to determine associations between structures of similar external morphology and function but differing in histological structure and developmental origin.

The symphyseal odontoid demonstrates two facets consistent with wear from occlusion with the anterior supragnathal odontoids positioned antero- and posterolabially (Figure 2A). Posterior to the symphyseal odontoid is the occlusal margin comprising the oral division, which is the shearing edge of the infragnathal. Between the symphyseal

odontoid and occlusal blade there is a wear incision, resulting from occlusion with the overlapping lateral odontoid of the anterior supragnathal. The size of this incision varies with age, becoming particularly large in terms of width and depth in adult and senescent individuals. The occlusal margin may bear one or two accessory odontoids. One, the intermediate odontoid, is almost always present and is located approximately midway along the anteroposterior length of the occlusal blade, whereas the other, the accessory odontoid, is positioned close to the wear incision for the anterior supragnathal odontoid and may be lost with extreme wear. At the posterior end of the occlusal blade is an elevated region resulting from wear here by the posterior supragnathal. This structure is called the posterior corner of the occlusal blade (not an odontoid).

The posterior, sloping margin of the posterior corner may bear a row of marginal teeth, although their presence is individually variable in *Dunkleosteus terrelli*. There is no clear correlation between size and the development of the marginal teeth, including such factors as wear or resorption with age. Small individuals may have no marginal teeth and the very largest infragnathal of *D. terrelli* (CMNH 5936) retains marginal teeth. Variation in the presence of tooth rows in other species of this genus is unknown, and whether these teeth are functional or merely vestigial in dunkleosteid arthrodires is also unclear. However, arthrodires from the Gogo Formation, including the dunkleosteid *Eastmanosteus*, show a large amount of variation in the number of teeth within tooth rows [51]. These non-shedding (statodont) teeth are demonstrated to be closely related to the formation of bone tissue in the basal growth zones of the gnathals. Wear patterns and comparison to other arthrodires suggest these teeth are added to the posterior end of the oral division and are not replaced in the same position during the animal's growth [20,46,52], being progressively worn and exposing the bony margin successively polished by lifetime interaction with the upper jaw elements [11]. All structures on the occlusal margin (except the marginal dentition) seem to be formed as a result of interaction between the gnathal elements of the lower and upper jaw, being regulated by their internal (histological) structure.

The lingual side of the oral division (Figure 2B) on the infragnathal bone can be divided into three main structures: the lingual fossa, the Meckelian contact area and an anterior process of the adductor division ("axial" component of Ørvig [46] and Rücklin et al. [52]). The latter was positioned between the lingual fossa and the area for Meckelian contact on the internal side of the jaw; its surface is smooth and very slightly concave. The lingual fossa extends to the symphyseal odontoid anteriorly, which posteriorly, with the posterior corner, sometimes forms a slight swelling overhanging this region. The lingual fossa is subdivided anteriorly and posteriorly into two regions by a buttress, extending to the base of the intermediate odontoid (Figure 2B). The surface of the lingual fossa is ornamented by resorption bays, or Howship's lacunae.

The morphology of the ventral margin of the infragnathal is mostly defined by its interaction with the supporting Meckelian cartilages (Figure 3). The Meckelian cartilages of arthrodires consist of three parts: an ossified articular in the mandibular joint, mentomandibular in the symphysis and an intermediate element that spans these two regions [10,11,53–55]. *Dunkleosteus* is one of the few pachyosteomorph arthrodires for which the mentomandibular element is known [10,54]. The only other pachyosteomorphs for which this element has been reported are the dunkleosteid *Eastmanosteus* [12] and the aspinothoracidans *Titanichthys* [54,56] and *Diplognathus* [57], and only in *Eastmanosteus* is the mentomandibular element complete. Miles [11] also postulated the presence of an unpaired cartilaginous basimandibular element linking the Meckelian cartilage antimeres, as exemplified by *Coccosteus cuspidatus*. Whether a basimandibular element was present in *Dunkleosteus* is unclear.

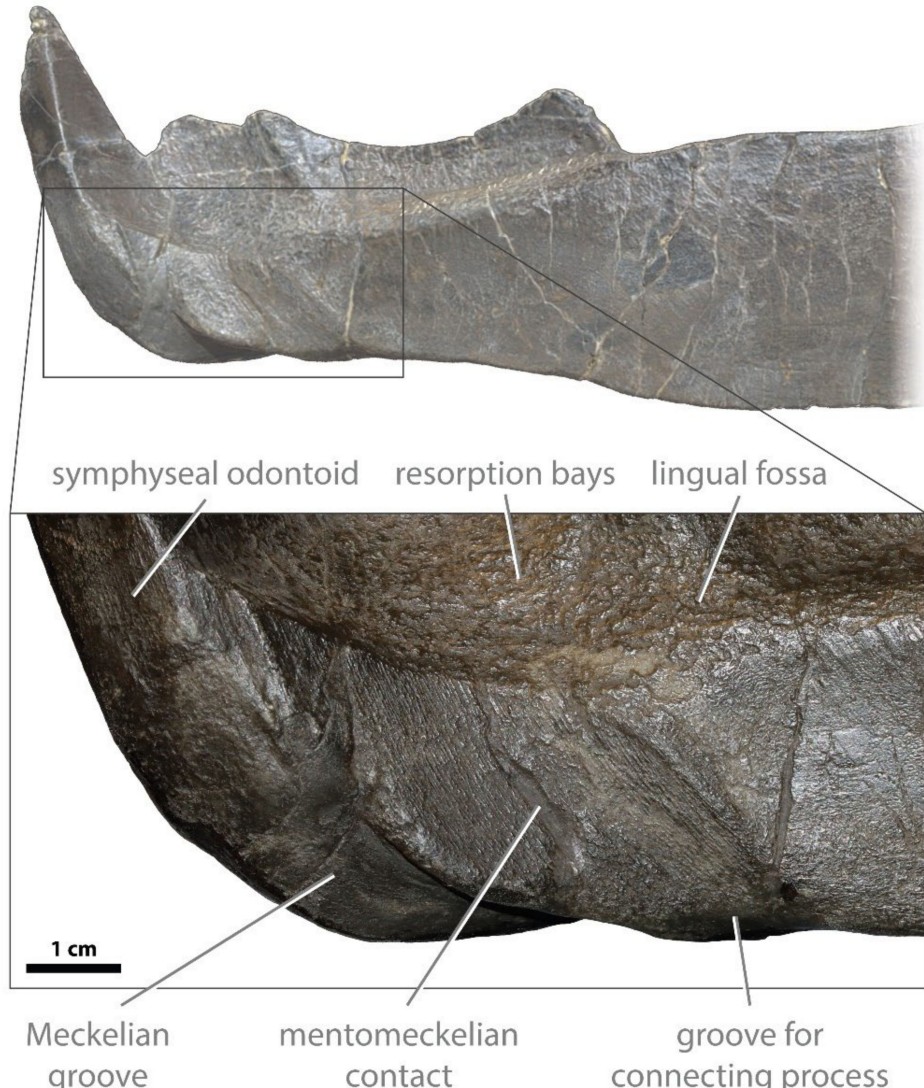

**Figure 3.** Morphology of the antero-ventral part of the infragnathal of *Dunkleosteus terrelli* (CMNH 7069) in lingual view.

The mentomandibular cartilage is known to be ossified and preserved only in some *Dunkleosteus terrelli* specimens (e.g., CMNH 6090, 7054 and 5768). In these specimens, the main, block-like element of the cartilage attaches to the mentomandibular contact positioned lingually on the infragnathal (Figures 2B and 3). Along with this, an additional, elongated part of the ventral Meckelian cartilage fits into a ventral groove that runs from the base of the symphyseal odontoid and connects to the mentomandibular cartilage via an overhanging lip of the connecting process. In those specimens in which the cartilage is not preserved, there is a wide and shallow groove at the posteroventral end of the mentomandibular contact, which housed the connecting process. Further posteriorly, the Meckelian cartilage emerges on the labial side of the infragnathal, where it meets the intermediate Meckel's cartilage (Figure 3). In those specimens in which the cartilage is not preserved, there is a wide and shallow groove (here termed the Meckelian groove, Figure 3) at the posteroventral end of the mentomandibular contact, which housed the lip-like process mentioned above. The surface of the mentomandibular contact bears numerous rough grooves and ridges subparallel to its margins that may serve to better anchor the mentomandibular element to the infragnathal.

Previous studies have reconstructed the mentomandibular element of *Dunkleosteus* as being slightly visible in external view [11,53]. However, examinations of 3D models of *Dunkleosteus* specimens with this element in situ find the element is preserved slightly rotated from its contact with the internal face of the infragnathal and the anterodorsal terminus of the Meckelian groove. Digitally manipulating the mentomandibular element back into proper articulation results in it perfectly filling the contact area and reaching the contact between the base of the symphyseal odontoid and the Meckelian groove. This also results in the mentomeckelian element not being visible in external view when the element is in life position, unlike all other arthrodires where the mentomeckelian element is known (including *Eastmanosteus*). This also results in the latter, internal face of the mentomandibular element more clearly facing the anatomical midline.

It is unclear how widely a Meckelian groove is distributed within Arthrodira, though it seems to be broadly distributed in pachyosteomorphs. However, it appears to be absent (or at least does not form an incised groove, versus the Meckelian cartilage simply shifting from the external to the interior face of the infragnathal) in *Plourdosteus* [46,49], and is weakly developed, if at all, in at least some specimens of *Incisoscutum* (NHMUK PV P 50946); it is also weakly developed in *Compagopiscis* (WAM 95.1.3; WAM 96.5.675) and *Harrytoombsia* (WAM 70.4.254), and strongly developed in *Torosteus pulchellus* (WAM 70.4.265, WAM 91.4.31, WAM 91.4.31) and *Latocamumurus* (WAM 86.9.699). In specimens from the Gogo Formation, this position can be confirmed, as the mentomandibular element is often attached to the Meckelian groove on at least one infragnathal plate. Among pachyosteomorphs, the presence of this groove can be confirmed in *Dunkleosteus* (every specimen examined, e.g., CMNH 7069), *Gorgonichthys* (CMNH 7129), *Bullerichthys* [58], *Eastmanosteus* ("*E.*" *calliaspis*; [12]), *Titanichthys* [56] and possibly *Diplognathus* [57]. It is also present in an undescribed "coccosteomorph" from the Givetian of the Silica Shale (UMMP VP 58097), and Miles and Westoll [55] describe it as present in *Coccosteus*. In *Heintzichthys*, the Meckelian groove is present but very shallow (CMNH 5648, CMNH 6025). By contrast, the groove is clearly absent in *Holdenius* (CMNH 8031), *Mylostoma* (CMNH 7256) and possibly some of the selenosteids (e.g., *Stenosteus*; [59]), and its presence is ambiguous (leaning towards absent) in *Bungartius* (CMNH 7573). Lelièvre [60] suggests a Meckelian groove characterizes all arthrodires and was secondarily lost in eubrachythoracids. However, these authors only included coccosteomorphs as representatives of Eubrachythoraci; they were aware of the Meckelian groove of *Dunkleosteus* but were uncertain of its homology with the state in more basal arthrodires. The presence and morphology of a Meckelian groove has not been considered in prior phylogenetic analyses of eubrachythoracids (e.g., [56,61]), but was considered in the broader arthrodiran phylogeny of Lelièvre [60]. The distribution of this character suggests it may be useful in resolving the relationships between coccosteomorphs and the pachyosteomorph clades.

The adductor region is blade-like in shape and shows a less complex morphology compared to the oral region (Figures 2 and 4). The intermediate Meckelian cartilage spans most of the adductor division. The former does not ossify, but the dorsal margin of its contact with the infragnathal is sometimes visible as a slightly rugose semilunar band on the labial surface of the adductor division (Figure 4). At the posterior end of the adductor division of the infragnathal, the intermediate Meckelian cartilage contacts the articular cartilage associated with the jaw joint, which is only occasionally preserved. In many specimens, the dorsal margin of the adductor region forms a distinct process which bulges dorsally from the body of the adductor lamina. The bone shows a roughened, irregular surface texture in the immediate vicinity of this process, suggesting the attachment of soft tissue and tentatively interpreted here as a part of the insertion area of the *m. adductor mandibulae* (Figure 4). This process is more distinct in smaller specimens than larger ones.

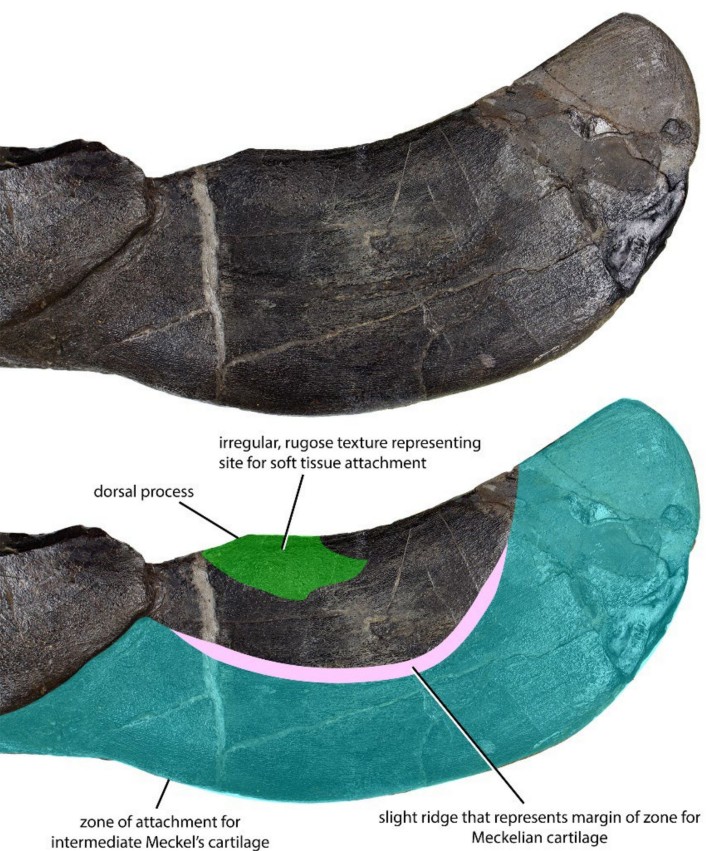

**Figure 4.** Left infragnathal of *Dunkleosteus terrelli* (CMNH 6194) in lateral view, showing band of connective tissue marking its contact with the intermediate Meckelian cartilage, the dorsal process, and the irregular, rugose texture at its base that indicates an attachment for soft tissue.

Denison [62] hypothesized the presence of two ossification centers in the arthrodire infragnathal, which Ørvig [46] later confirmed in *Plourdosteus canadensis.* Ørvig [46] suggested that during ontogeny, these regions became fused but stayed separated by a boundary still seen in cross-sections of adult infragnathals. Ørvig [46] ossification centers include an 'axial' component composed of the elongate adductor lamina posteriorly, plus its anterior, internal projection on the internal surface, and a 'dental' component, composed of the oral division and its teeth. However, the former does not exactly follow the jaw axis and the latter does not bear teeth in all arthrodires (e.g., *Dunkleosteus*). Rücklin et al. [52] referred to the first of these terms as 'shaft', although it is not uncommon for other authors to name the outer component an adductor lamina or "blade". For this reason, we propose renaming these components as para-articular and oral, respectively.

### *4.2. Histological Studies*
#### 4.2.1. Previous Histological Studies of Gnathal Elements in the Dunkleosteidae

Despite the Dunkleosteidae being studied for more than 150 years [7,8], little attention has been paid to the histology of their gnathal bones. The earliest description of the gnathal histology of *Dunkleosteus terrelli* was presented by Claypole [18], who was the first to state the bony tissue of the gnathal elements does not differ in structure from the other bones of this fish, with the exception of the inner part of the infragnathal which was much harder and denser. He noted the vascular canals were very thin and scarce in this bone, and found no dentine tissue. Of special note is his remark on the differentiation of bone density on the occlusal margin allowing for possible self-sharpening during the process of wear.

Hussakof [63] found no sign of a defined boundary between the oral and adductor divisions, the two being completely fused to each other.

Stetson [19] mainly repeated previous examinations of the infragnathals of *Dunkleosteus* ("*Dinichthys*") *terrelli* and *D. "intermedius"* (=*D. terrelli*; [64]), but also noted the apex of the symphyseal odontoid is composed entirely of dense bone, and the vascular (Haversian) canals are almost completely infilled, close to the actual occlusal surface of the bone. Stetson [19] also noticed a sharply defined boundary between the cancellous and compact bone types in the symphyseal odontoid.

Heintz [10] presented thin sections of the posterior supragnathals and the symphyseal odontoid ('pick') of "*Dinichthys*" (*Dunkleosteus terrelli*). Comparisons of the microstructure of those elements to that of the interolateral bone of the trunkshield led him to conclude there was no difference in histology between the gnathal and armor bone, apart from the former exhibiting more compact bone with narrower Haversian canals and more massive and dense lamellae.

Johanson and Smith [20,65] focused on tooth structure, distribution and development in various placoderms. These authors found the gnathal teeth of coccosteomorphs are composed of regular dentine rather than semidentine, as previously suggested by Ørvig [46]. They suggested the intermediate odontoid ('main tusk') is formed of an ingrowing vertical column of pleromic dentine rather than semidentine, as suggested earlier by Ørvig [46], and may be derived from the row of regular marginal teeth. Additionally, Johanson and Smith [20,65] described a process of active remodeling by successive resorption and redeposition of bone and dentine on the lingual side of the infragnathal in the coccosteomorph arthrodire *Incisoscutum*. In some cases, migratory dentine cells may have invaded soft tissue spaces in the gnathal elements to produce pleromic dentine, comparable to the ingrowing of osteonal bone in the occlusal margins of dunkleosteids, compensating for tissue loss by wear, although these cases are very rare in placoderms. Johanson and Smith [20] presented sections of the gnathal bone of *Dunkleosteus* sp., demonstrating new osteons filling in vascular spaces, but no dentine invasion had been recorded on their material.

### 4.2.2. Histology of the Gnathal Elements in Dunkleosteidae Indet

To interpret results obtained from micro-CT studies of the holotype of *Dunkleosteus tuderensis* Lebedev sp. nov., we selected a number of fragmentary infragnathal and supragnathal specimens to prepare thin sections (Figures 5–10). Because only a single dunkleosteid specimen has been reported from the type locality, those specimens originating from a roughly synchronous locality, Gornostayevka, in the Orel Region of Russia, were used.

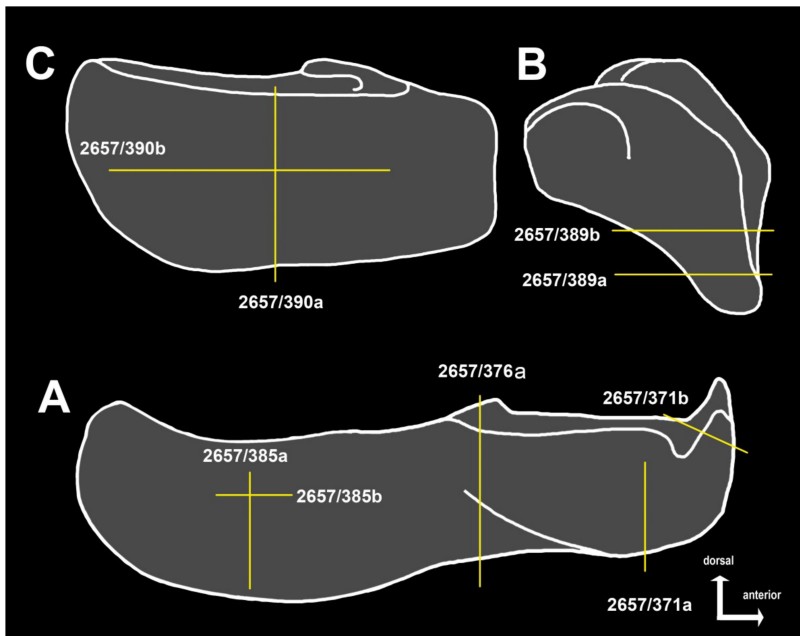

**Figure 5.** (**A**–**C**), diagrams explaining relative positions of sections taken for histological examination.

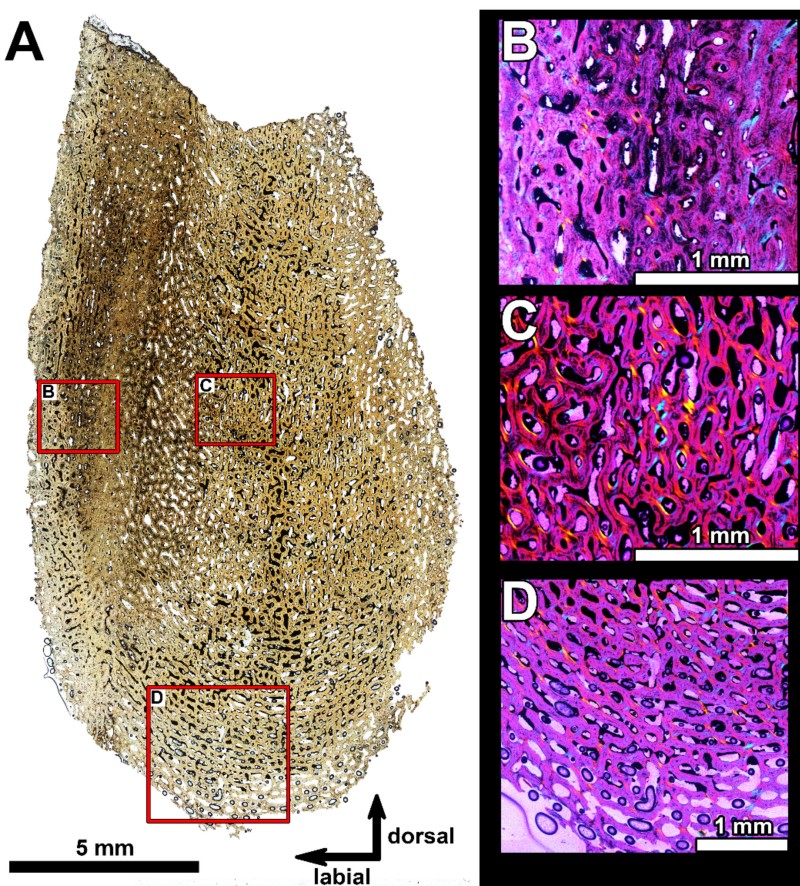

**Figure 6.** (**A**), Histological section of the ventral part of the oral division of the infragnathal PIN 2657/371 (section PIN 2657/371a, Figure 5), showing microstructural overview under normal light. (**B–D**) are enlarged areas photographed under polarized light with lambda waveplate showing highly vascularized bone tissue lacking diploë structure. Note deposition of circumvascular (fine-fibered) bone on the vascular canal walls.

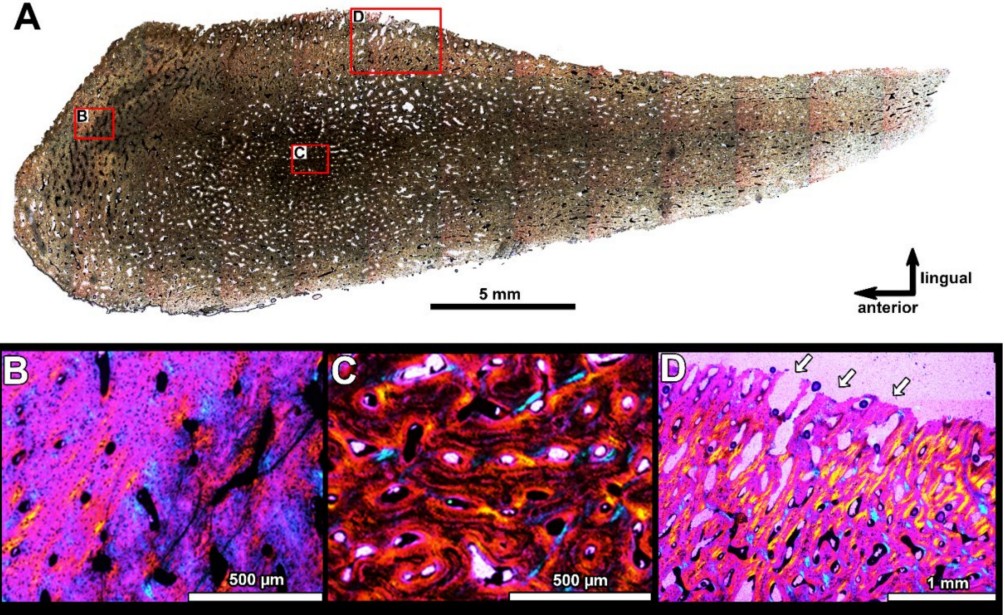

**Figure 7.** (**A**), Histological section at the base of the symphyseal odontoid in the infragnathal PIN 2657/371 (section PIN 2657/371b, Figure 5), showing microstructural overview under normal light.

(**B**–**D**) show details of the vascular canals structure under polarized light with lambda waveplate. Note deposition of lamellar bone on the walls of vascular canals in the central part (**C**) and superficial resorption marked by white arrows on the lingual side (**D**). Several of the osteons in (**C**,**D**), seen in polarized light, reveal the circular arrangement of crystal fiber bundles in this tissue around the blood vessel, through the opposite sign of birefringence (blue and yellow with a gypsum plate) known as a Maltese cross, as in any type of this mature circumvascular deposition of layers.

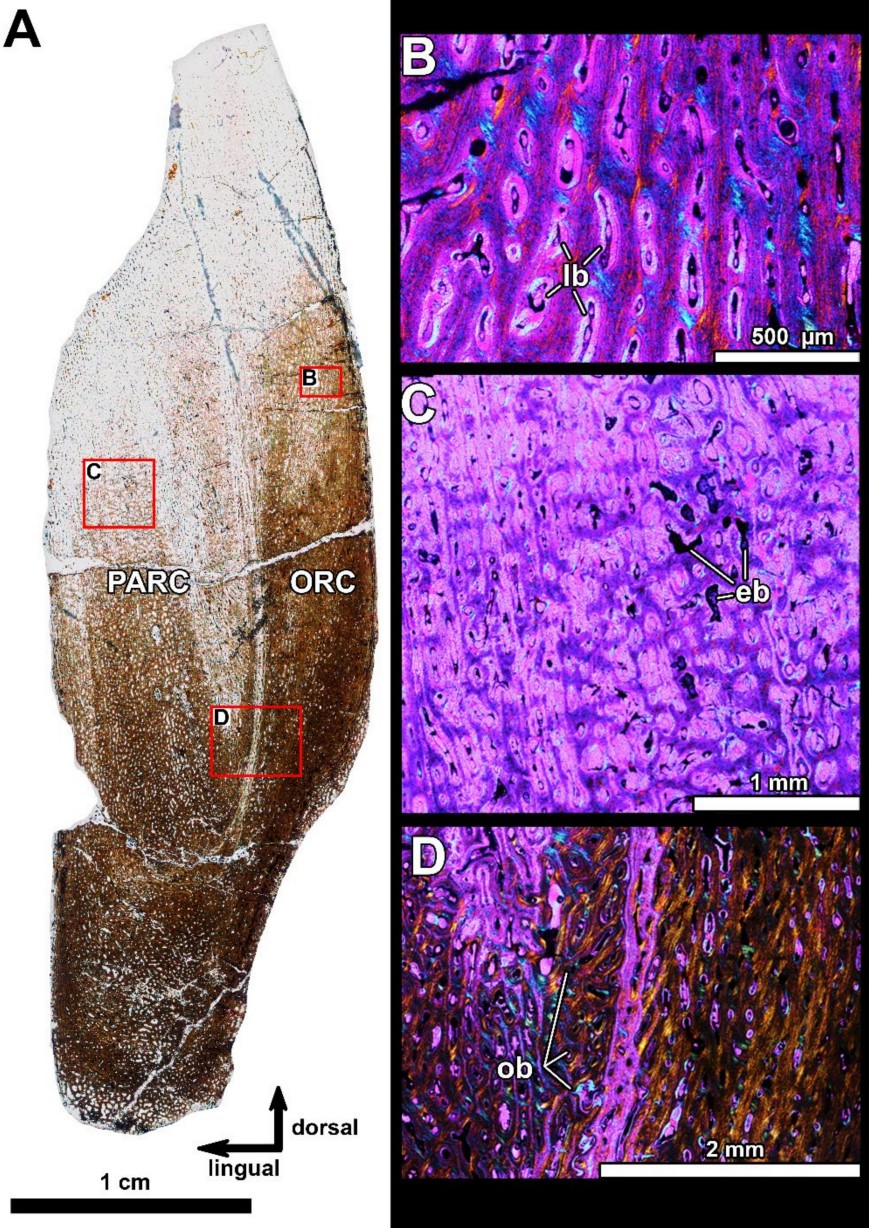

**Figure 8.** Histological section in the posterior part of the oral division of the infragnathal PIN 2657/376 (section PIN 2657/376a) showing microstructural overview under polarized light with lambda waveplate (**A**), and details showing interrelationships of the para-articular and oral components of the infragnathal (**B**–**D**). Note the presence of the lamellar type of vascularization characteristic of fast-growing primary bone (**B**) and the osteonal bone in the para-articular component along the boundary with oral component (**D**). Abbreviations: eb, erosion bays; lb, lamellar bone; ob, osteonal bone; ORC, oral component; PARC, para-articular component.

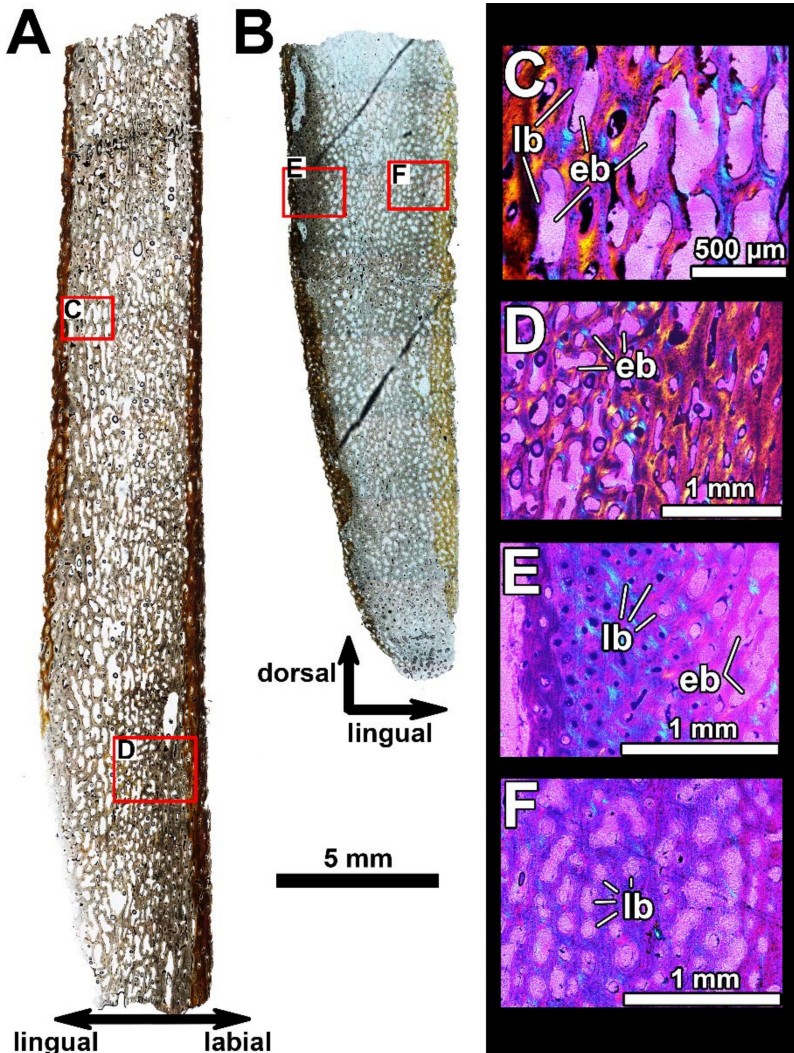

**Figure 9.** Histological sections of the adductor lamina of the infragnathal PIN 2657/385 (sections PIN 2657/385a, b), showing microstructural overview in the longitudinal horizontal plane (**A**) and in the transverse vertical plane (**B**) under normal light, showing the presence of diploë structure. (**C**–**F**) show details of the vascular canal structure under polarized light with lambda waveplate; (**C**) shows examples of Maltese cross secondary osteons. Note various sized erosion bays in the middle of the cancellous layer. Abbreviations: eb, erosion bays; lb, lamellar bone.

Some difficulties regarding the identification of several histological features were encountered due to fact that histological sections could not be taken for an ontogentic series of gnathal bones, tracing the development of the bone structure. Thus, our study is based on the histology of bones of presumably adult or sub-adult animals. The interpretation of how vascular canals are modified during ontogeny is also problematic. The primary position of vascular canals in the studied gnathals was easily observable in all sections and had not changed during ontogeny. However, primary vascular canals might have changed in two ways: first, either being filled with deposited osteonal circumvascular bone without prior resorption (formation of primary osteons without resting lines), or secondly, local resorption of primary vascular canals or primary osteons accompanied by expansion and further filling by lamellar bone (formation of secondary osteons with characteristic resorption lines at the border of new deposition). Secondary osteons disrupt the primary bone structure, including the position of the primary vascular canals; this is not observed in our sections (Figures 6–11). On the other hand, some vascular canals are clearly enlarged due to local resorption (e.g., eb, Figures 9–11) and circumvascular

bone is deposited on their eroded walls (Figures 8B, 9 and 10), being consistent with the definition of secondary osteons, although the resorption surface, or resting lines producing the distinctive cross-cutting relationships across primary vascular canals or primary osteons often cannot be easily found. Thus, we cannot reliably trace the pathways of ontogenetic changes of vascular canals on the available material, and simply term the modified vascular canals 'osteons'. In accordance with previous studies [20], we term the dense bone tissue formed by osteons as 'osteonal bone' and put aside the question of the origin of these osteons (primary or secondary) for future histological studies of ontogenetic series of gnathal bones.

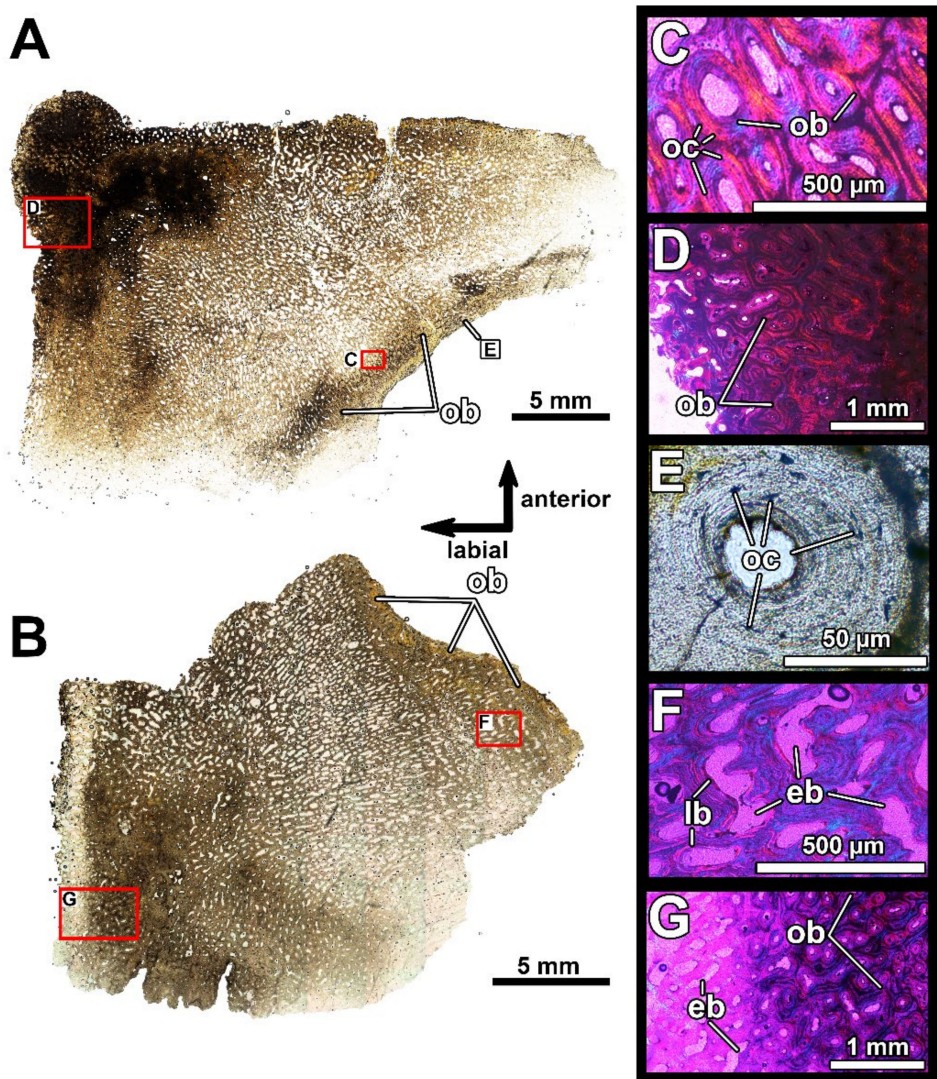

**Figure 10.** Transverse histological sections of the odontoid of the anterior supragnathal PIN 2657/389 parallel to each other; section PIN 2657/389a is cut more apically (**A**), section PIN 2657/389b is closer to the bone base ((**B**); Figure 5). (**A**,**B**) are microstructural overviews photographed under normal light and showing highly vascularized bone tissue and presence of osteonal bone along its occlusal margins. Structural details of vascular canals photographed under polarized light with lambda waveplate are shown in (**C**,**D**,**F**,**G**). The osteon (**E**) of the osteonal bone under normal light showing the presence of osteocyte lacunae. Note the presence of osteonal bone (**C**,**D**,**F**,**G**), erosion bays in the cancellous inner part (**F**) and transition between cancellous bone with erosion bays and osteonal bone (**E**,**G**). Abbreviations: eb, erosion bay; lb, lamellar bone; ob, osteonal bone; oc, osteocyte lacunae.

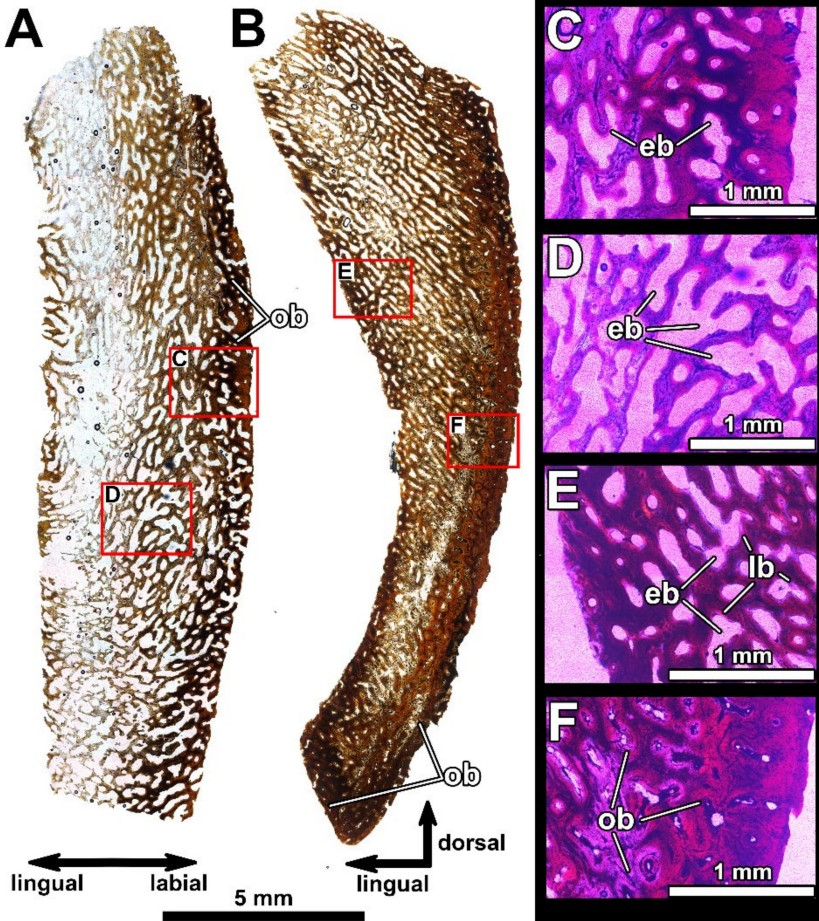

**Figure 11.** Histological sections of the posterior supragnathal PIN 2657/390 (sections PIN 2657/390a, b), showing microstructural overview in the longitudinal horizontal plane (**A**) and in transverse vertical plane (**B**) under normal light, showing cancellous structure and presence of osteonal bone along the occlusal margin. (**C–F**) present structural details of the vascular canals photographed under polarized light with lambda waveplate. Note the presence of erosion bays forming the cancellous structure of the bone and the osteonal bone along the occlusal margin. Abbreviations: eb, erosion bay; lb, lamellar bone; ob, osteonal bone.

Despite these difficulties, many histological features may be reliably identified. The sections (Figures 6–11, respectively) are broadly similar, consisting of highly vascularized bone. The primary bone organization is very similar to that in the tetrapod fibrolamellar complex (sensu [66]). The gnathal bones, in most of their parts, lack a typical distinct diploë structure—a basic pattern of organization in most of the dermal bones, including three distinct layers: an external cortex, a middle cancellous region and a compact internal cortex. The diploë structure can be found only in the thin, flat posterior part of the infragnathal (=adductor lamina). There is no trace of any dentine tissues (including semidentine and/or pleromic dentine). The superficial resorption may be traced on the lingual side of the infragnathal as a sharp line disrupting the internal structure of the bone, for example, the vascular canal network (Figure 7D, arrows).

Bone remodeling (namely erosion pits bays and formation of osteons) is active in the gnathal bones. It is extensive and most pronounced in the areas subject to increased functional stress, for example, the odontoid on the anterior supragnathal and the occlusal margin of the posterior supragnathal (Figures 10 and 11). Active bone remodeling results in a transformation of vascular canals and a rise of numerous osteons along the functional wear facets. In these areas, the osteons form a dense bone tissue, an 'osteonal bone' (sensu [20]). The osteonal bone is also present along the labial face of the para-articular component.

The ventral part of the anterior portion of the infragnathal PIN 2657/371 (transverse thin section PIN 2657/371a (Figure 5), posteriorly from the symphyseal odontoid, Figure 6) is entirely composed of relatively homogeneous primary bone tissue and lacks any trace of a diploë structure. The primary bone is highly vascularized and osteons are arranged in circumferential rows. The orientation of vascular canals is predominantly longitudinal, but reticular canals are also present. The number of canals remains unchanged closer to the bone surface. Bone remodeling is restricted to an alteration of some primary vascular canals into erosion bays that generally replicate their form and the deposition of lamellar bone on the walls of vascular canals.

The histological structure of bone tissue at the base of the symphyseal odontoid and an area immediately behind it (oblique horizontal thin section PIN 2657/371b, Figure 6) is generally similar to that described before for the anterior part of the infragnathal, for example, in the presence of highly vascularized primary bone and the lack of a diploë structure. The bone is least compact along the symphyseal and lingual margins due to the presence of reticular and radial vascular canals. The transition between the areas of compact and less compact bone is gradual. It can be observed that one surface is resorptive and the other displays bone deposition that would suggest selective loss linked with definitive growth, the way bones can increase in directional size.

Bone structure in the posterior part of the occlusal margin of the infragnathal (transverse vertical section PIN 2657/376, Figure 8) is generally similar to that found in the anterior part of the infragnathal but demonstrates that the para-articular and oral bone components were separated by a distinct boundary. The microanatomical and histological structures of these bone components are generally similar, although the osteonal bone in the para-articular component is more compact (Figure 8D). Vascular canals are mostly longitudinal in the para-articular component and randomly oriented in the oral one, including the lamellar type of vascularization (Figure 8B) characteristic of fast-growing primary bone. The number of vascular canals does not decrease towards the bone surface of the oral component, and numerous canals open to the surface indicating that bone growth continued in this direction.

The thin and flat adductor lamina of the infragnathal (PIN 2657/385a, b, Figure 9) demonstrates a three-layered diploë structure due to the presence of a middle cancellous region. This is spongy, being riddled with cavities representing erosion bays of various sizes.

The transverse horizontal sections of the odontoid of the anterior supragnathal PIN 2657/389a, b (Figure 10) show highly vascularized bone tissue. The inner part is cancellous and contains numerous erosion bays formed in the place of the modified vascular canals. Extensive bone remodeling also results in the emergence of numerous osteons along the functional margin and the formation of dense osteonal bone. Figure 9C, E is a good example of the circular arrangement of the deposition of circumvascular bone and the Maltese cross arrangement.

In the posterior supragnathal specimen, sections PIN 2657/390a, b (Figure 11) show a cancellous structure due to the presence of elongated, regularly arranged erosion bays formed in the vascular canal area. The areas along the labial surface and the occlusal margin are compact due to the presence of dense osteonal bone.

Both supragnathals are uniform in their internal structure and include no separate bone components.

### 4.3. Systematic Status of KMA 4155

Within the Dunkleosteidae, infragnathals have been reported only for the genera *Dunkleosteus*, *Golshanichthys*, *Eastmanosteus* and *Hussakofia*. The infragnathal structure of *Eastmanosteus* and *Hussakofia* strongly differs from that seen in *Dunkleosteus* and *Golshanichthys* [12,22]. *Eastmanosteus* has a highly denticulate occlusal margin and well-developed symphyseal teeth [12], whereas in *Hussakofia,* the infragnathal is very short and the oral region is slightly curved [67], ruling out the assignment of KMA 4155 to these taxa. The infragnathal of *Golshanichthys* is proportionally much deeper than that of KMA

4155. Additionally, its symphyseal odontoid is much lower and the intermediate odontoid is well expressed, as is the buttress supporting it and dividing the lingual fossa into two sections, as in *Dunkleosteus*. However, in *Golshanichthys*, the anterior fossa is larger than the posterior one, whilst in *D. terrelli*, this proportion is reversed [68].

Specimen KMA 4155 also resembles the aspinothoracidan *Holdenius holdeni* [69] in some respects, namely in the almost straight occlusal margin, the lack of any accessory odontoids on the occlusal margin and a poorly developed to absent intermediate buttress. However, the oral division of the infragnathal of *H. holdeni* is proportionally longer, and the posterior part of the occlusal margin is elevated relative to the anterior part and bears a row of lingually directed teeth.

Thus, KMA 4155 most closely resembles members of the genus *Dunkleosteus* Lehman, 1956. However, the taxonomy of this genus itself is far from perfect. *Dunkleosteus* currently exists as a wastebasket taxon containing a large number of species established upon various isolated, non-overlapping skeletal elements, and many are insufficiently characterized or probably synonymous with other taxa. This is also not considering additional *Dunkleosteus* material from California [70], Texas [71] and Poland [72] which has been only referred to as *Dunkleosteus* sp.

Of the various taxa currently assigned to *Dunkleosteus*, only *D. terrelli* and *D. raveri* are taxonomically stable. *D. belgicus*, *D. missouriensis*, *D. denisoni* and *D. amblyodoratus* are all based on isolated plates or highly fragmentary material, diagnosed by a small number of characters, while the distinctiveness of these characters relative to individual variation within the large *D. terrelli* hypodigm is uncertain [22]. *D. belgicus* has been suggested to be a nomen dubium and the material assigned to it may pertain to the genus *Ardennosteus* [73]. Some of these taxa (specifically *D. amblydoratus*) may prove to be valid, but many appear to be topotypic names for *Dunkleosteus* material from Belgium, Missouri, Poland and the Kettle Point Formation of Ontario, respectively. *D. raveri* is also diagnosed based on a single character (a tuberculate skull roof; [5]), but this is a character that shows less individual variation in arthrodires. However, populations of *Plourdosteus trautscholdii* have been reported in which individuals with tuberculate and non-tuberculate armor coexist (D. Goujet in [5]: p. 204), and in Gogo Formation arthrodires, tuberculation is often reduced and restricted to the margins of plates in older individuals [51].

The Frasnian *Dunkleosteus magnificus* and *D. newberryi* are known from more extensive material and are probably valid species (see [74]), but whether they belong to the genus *Dunkleosteus* or even the Dunkleosteidae is unclear. Both Schultze [25] and Dennis-Bryan [12] suggested *D. magnificus* may pertain to a very large (*D. terrelli*-sized; estimated length ~3 m based on a total cranial length of 55 cm and comparisons with CMNH 6090, 7054 and 5768; [1,74]) representative of *Eastmanosteus*. Gnathals are known for both species but could not be examined directly. The holotype of *D. magnificus* appears to have a much more prong-like accessory odontoid than any other species of *Dunkleosteus* ([74]: pl. 7.1), but it is unclear if this is an individually variable trait. *D. newberryi* has a very long tooth row compared to the blade region and seemingly no accessory odontoids ([74]: pl. 10.2). However, both of these taxa are in need of redescription and placement in a phylogenetic analysis.

The status of *Dunkleosteus marsaisi* is also uncertain. Near-complete specimens of *D. marsaisi* are known in museums (e.g., the Paläontologische Sammlung der Universität Tübingen) and private collections, but little has been published on this species since its initial description by Lehman [75]. Some authors have even suggested *D. marsaisi* merely represents a North African topotype of *D. terrelli* [76]. Specifically, for the purposes of this study, limited data are available on the gnathals of *D. marsaisi* for comparison. The gnathals of the holotype of *D. marsaisi* are not figured in Lehman [75] and photographs of them show these elements are poorly preserved ([77]; also photos courtesy of A. Pradel), making comparison deficient.

The most prominent feature of the infragnathal in the holotype of *Dunkleosteus marsaisi* is a very slender and short buttress of the intermediate odontoid. Although its apex is completely broken off, it may be seen that the base of the buttress is significantly weaker

than in most *D. terrelli*, with the base of the buttress in *D. marsaisi* not extending to the bottom of the lingual fossa. This resembles the condition in specimen KMA 4155, although it is still better expressed in the former specimen. Strong widening of the area of symphyseal contact in *D. marsaisi* results in the contact with the mentomandibular shifting posteriorly and the anterior portion of the lingual fossa being strongly reduced.

This chaos in the taxonomy of *Dunkleosteus* is largely inherited from the late 19th century, when *Dunkleosteus* was "*Dinichthys*" (sensu lato) and nearly every arthrodire with symphyseal odontoids and blade-like oral margins was referred to this taxon (see discussion in [5,10,22,78]). Some of these species have since been split off into the genera *Dinichthys*, *Gorgonichthys*, *Eastmanosteus*, *Heintzichthys* and *Hussakofia*, but a large number have yet to be revised. *Dinichthys herzeri* was retained in its own monotypic genus [5,75], but the rest of "*Dinichthys*" (sensu lato) remained with *Dunkleosteus*.

Thus, the systematic attribution of new material to *Dunkleosteus* and whether it pertains to a species distinct from *D. terrelli* is largely dependent on the intraspecific variability of infragnathal morphology in *Dunkleosteus terrelli* (which, aside from Hlavin [64], has never been examined), and comparisons with the only known, poorly preserved infragnathal of *D. marsaisi*.

### 4.4. Intraspecific Variation in the Oral Division of Dunkleosteus terrelli

*Dunkleosteus terrelli* specimens can be broadly separated into two morphotypes (Figure 12A,B), hereafter termed Morphotype A and Morphotype B. These two groups differ significantly in size, with morphotype A containing almost exclusively smaller individuals of *Dunkleosteus* with infragnathal lengths of 18–37 cm and estimated total lengths of 1.5–2.4 m. Morphotype B includes larger specimens with infragnathal lengths of 35–70 cm and estimated total lengths of 2.4–4.0 m (using method of [1]). In Morphotype A (Figure 12A), there are two accessory odontoids on the occlusal margin, with an accessory odontoid between the symphyseal and the intermediate odontoid. The notch between the symphyseal odontoid and occlusal margin is also very narrow. In Morphotype B (Figure 12B), the accessory odontoid has been obliterated through wear and the notch between the symphyseal odontoid and occlusal blade is much broader and more open, most likely being correlated with a larger specimen size (= older age of the animal?). The posterior tooth row of Morphotype B is typically longer and oriented at a much shallower angle than that of Morphotype A. The symphyseal odontoids in Morphotype A are somewhat taller than those in the second one, irrespective of size. However, specimens in Morphotype B tend to have proportionally larger symphyseal odontoid bases.

In Morphotype A, the anterior lingual fossa, measured from the buttress to the symphyseal margin, is roughly 2/3 the length of the posterior fossa, which is measured to the posterior end of the oral region. This results in a ratio between these two regions of 1:1.5. However, in Morphotype B, the posterior region is longer, with a ratio of approximately 1:2. This may be due to ontogeny; in other arthrodires, the oral region of the dentary expands through the addition of teeth at the posterior end of the existing structure [52], and if the position of the intermediate buttress remained constant, this would result in the posterior region becoming longer with growth. This also fits with previous suggestions that the jaw of *Dunkleosteus* became proportionally longer with age [1,10,79].

Variation in these other features appears to be due to ontogeny, as they do not covary in a way that consistently diagnoses morphological groups but appear to be correlated with size (see also [64]). However, the possibility cannot be entirely ruled out that these morphotypes correspond to two highly similar species primarily distinguished by size, similar to extant *Carcharhinus* spp. (e.g., [80,81]). Other features, such as the development of the intermediate buttress, the odontoid's depth and angle of inclination relative to the longitudinal axis of the jaw and the inclination of the symphyseal odontoid, are also individually variable.

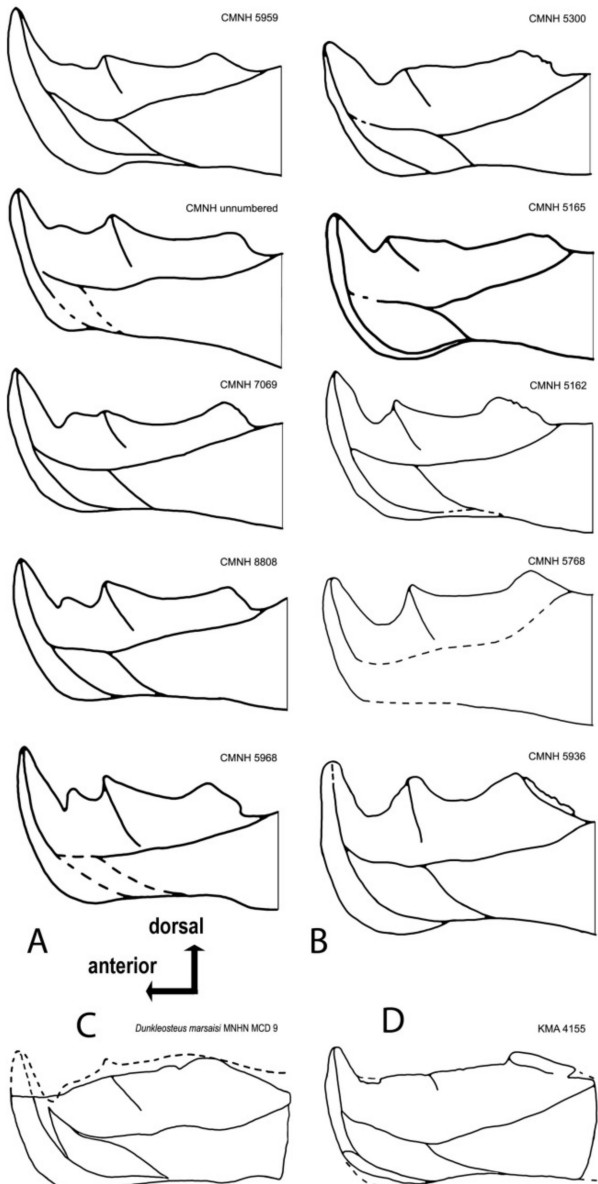

**Figure 12.** Variations of morphological structure seen on the lingual side of the oral division of the infragnathal in *Dunkleosteus terrelli* separated into two groups: (**A**) including smaller and (**B**) larger individuals. (**C**) is an outline drawing from a photo of a specimen MNHN MCD 9; (**D**) an outline drawing from the holotype of *D. tuderensis* Lebedev, sp. nov. All left jaw specimens mirrored for easier comparison. (**C**) drawn after a photo courtesy of A. Pradel (MNHN, Paris, France).

In order to determine whether the structure of KMA 4155 fits within the range of variation seen in *Dunkleosteus terrelli* infragnathals, we applied a graphical landmark analysis (Figure 13). Nine landmarks were used for the labial projection of the oral division of the infragnathal: (1) base of the anterior labial facet of the symphyseal odontoid, (2) apex of the symphyseal odontoid, (3) ventral apex of incision of anterior supragnathal odontoid, (4) anterior corner of the occlusal margin, (5) apex of the intermediate odontoid, (6) posterior corner of the occlusal margin, (7) deepest point of the notch formed by the occlusal blade and the adductor lamina, (8) ventral point of the contact between the oral division and the adductor lamina labially, (9) extreme projection of the ventral point of the symphyseal margin and six more for the lingual side, (10) dorsal apex of incision of anterior supragnathal odontoid, (11) base of intermediate odontoid, and (12)–(15) landmarks delimiting the contact area for the mentomandibular cartilage.

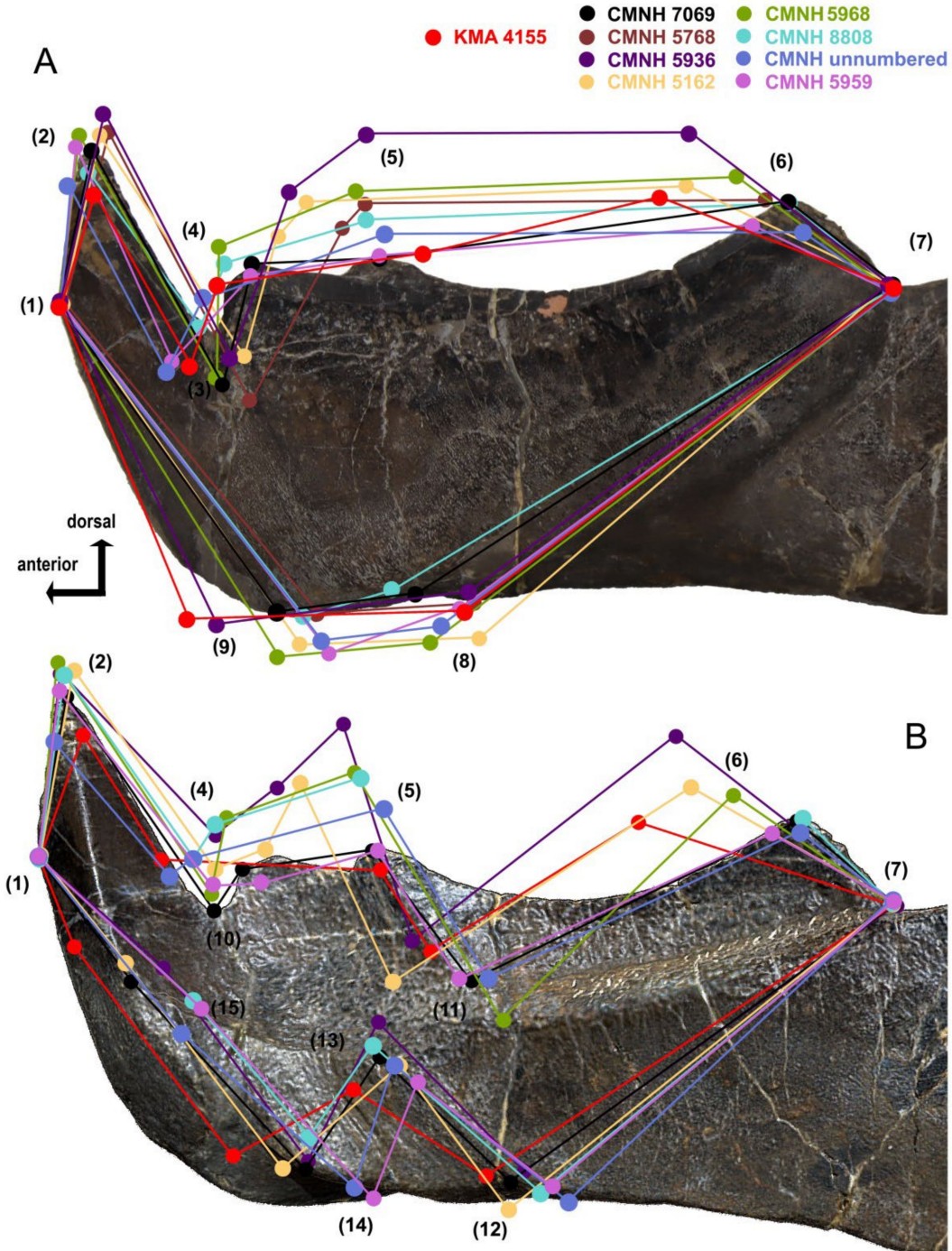

**Figure 13.** Graphical landmark analysis of the structure of the oral division of the left infragnathal in several specimens of *Dunkleosteus terrelli* (CMNH 7069 used as a basis) in comparison to *D. tuderensis* Lebedev, sp. nov. from the labial (**A**) and the lingual (**B**) sides. See landmarks position (1–15) in the text. All jaw specimens mirrored to be in the same orientation for easier comparison.

CMNH 7069, which is close to the average size and morphology for the entire sample of *D. terrelli*, was used as a basis upon which to compare landmark distribution from the other infragnathals (Figure 13). The distance between (1) and (7) was taken as a constant, and for this reason, excluded from the analysis. Landmark configurations were brought to the same size in a graphic processing program and optimally aligned. The resulting dispersion of other landmarks resulting from variability in size and structure of individual specimens was analyzed.

Specimen CMNH 5936 is the greatest outlier in morphology in this dataset, but this is likely because this specimen is unusually large relative to other specimens of *Dunkleosteus* [1]. CMNH 5936 demonstrates a strong shift in the positions of landmarks (4)–(6), defining the shape of the occlusal blade and position of the corresponding intermediate odontoid apices, as well as the anterior shift of (9) (maximal projection of the ventral point of the symphyseal margin). Less significant deviations are demonstrated by CMNH 5162, in which the posterior angle of the occlusal margin is shifted anteriorly (6). The other specimens demonstrate no significant deviations. Some difference in position between the labial and lingual projections may be due to a slightly different orientation of specimens during photography.

This examination suggests that the most variable features in the structure of the oral division of *Dunkleosteus terrelli* are the development and position of the intermediate odontoid (5), the development and position of the posterior corner of the occlusal blade which may or may not carry a tooth row, and variations in the projection of the ventral point of the symphyseal margin (9). Most of these deviations can be attributed to the highly dynamic occlusal margin of arthrodires, in which the oral region is continually remodeled throughout life due to growth and wear thegosis [82] against the interacting supragnathal elements.

Most of the analyzed landmarks of KMA 4155 are within the range of variation seen in specimens of *Dunkleosteus terrelli*, with the exception of the extremely anterior projection of the ventral point of the symphyseal margin (9), the small size of the intermediate odontoid (5) and the very shallow posterior corner of the occlusal margin (6). In KMA 4155, (6) is oriented at a much shallower angle than in specimens of Morphotype A of *D. terrelli*. Instead, in the very largest specimens of *D. terrelli* (Morphotype B; CMNH 5936), the posterior corner of the occlusal margin is shifted antero-dorsally. By contrast, landmark (5) is shifted posteriorly, reflecting, in sum with (6), a posterior location of the intermediate odontoid. Landmarks (2) and (14) in KMA 4155 occupy an extreme position, but are still within or close to the main dispersion cloud, at least within the limits of the range in variability presented by deviating characters in some specimens of *D. terrelli* described above. This suggests KMA 4155 represents a form close to *D. terrelli*, but differs enough to be recognized as a new species.

## 5. Systematic Paleontology

<div align="center">

Placodermi McCoy, 1848

Arthrodira Woodward, 1891

Pachyosteomorphi Stensiö, 1944

Superfamily Dunkleosteoidea Vézina, 1990

Family Dunkleosteidae Stensiö, 1963

Genus *Dunkleosteus* Lehman, 1956

*Dunkleosteus tuderensis* Lebedev, sp. nov.

</div>

Etymology. The specific epithet *tuderensis* refers to the locality of the discovery of the holotype specimen on the bank of the Maliy (Minor) Tuder River in the Tver Region of Russia.

Holotype. KMA 4155, oral division of a left infragnathal.

Provenance. Left bank of Maliy Tuder River about 200 m to the north–northwest of the currently abandoned Bilovo village, Toropets District, Tver Region, north-western Russia; Bilovo Formation, (?) Lower-Middle Famennian, Upper Devonian.

Diagnosis. Symphyseal odontoid short, its height three times smaller than greatest depth of the oral division from the base of the odontoid. Axis of symphyseal odontoid perpendicular to the occlusal margin. Occlusal margin almost straight, with posterior corner gently elevated. Intermediate odontoid poorly developed (vestigial), its buttress hardly discernible. Lingual shelf of the lingual fossa low, the anterior and posterior fossae grade smoothly into one another.

Remarks. Characters related to the structure of the occlusal margin are important because, despite their variability in *Dunkleosteus terrelli*, they reflect the structure of the anterior and posterior supragnathal elements and interaction between the upper and lower jaw elements, reflected in a number of characters.

Description (Figures 14 and 15; Supplementary Figure S2, in greyscale, to present the specimen in higher resolution and contrast).

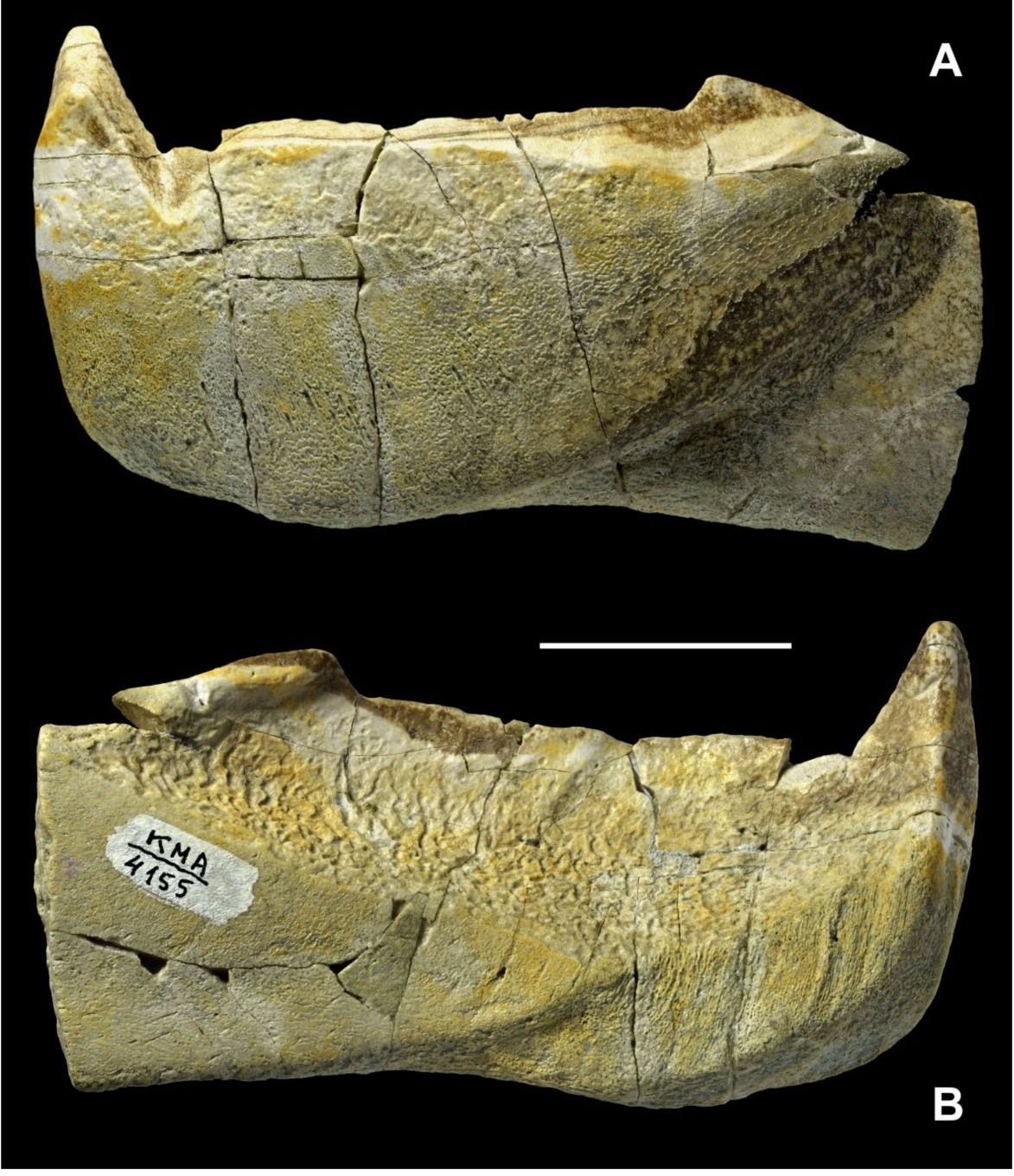

**Figure 14.** *Dunkleosteus tuderensis* Lebedev, sp. nov., holotype KMA 4155 in (**A**) labial and (**B**) lingual views. Scale bar = 3 cm.

### 5.1. General Morphology

The symphyseal odontoid is quadrangular but not quadrate in section, its symphyseal margin being almost parallel to the posterior labial facet (Figure 15A). The anterior labial

facet is oriented at an angle to the opposing lingual face of the odontoid. The maximum height of the symphyseal odontoid measured from its apex to the occlusal margin is three times larger than the greatest depth of the oral division and is equal to the horizontal distance between the ventral apex of incision of the anterior supragnathal odontoid and the symphyseal margin. This is much shorter than in the specimen of *Dunkleosteus terrelli* which shows the least development of the symphyseal odontoid (CMNH 5959). The symphyseal odontoid points strictly dorsally, with its longest axis being perpendicular to the rest of the occlusal margin. The incision of the anterior supragnathal odontoid is expressed only from the labial side, without forming a notch in the occlusal margin usually seen from the lingual side in *D. terrelli*, although a piece of bone is missing from the occlusal margin in this area. The longitudinal occlusal facet running along the occlusal margin widens posteriorly, reaching its maximum at the apex of the posterior corner (Figures 14A and 15B,C). The posterior margin of the posterior corner is smooth; it bears no teeth and is inclined to the main part of the occlusal margin at an angle of 27°. The ventral half of the labial surface of the occlusal division bears numerous thin parallel vascular grooves directed antero-dorsally. Those are most probably revealed by abrasion which removed a thin superficial bone layer originally covering the canals.

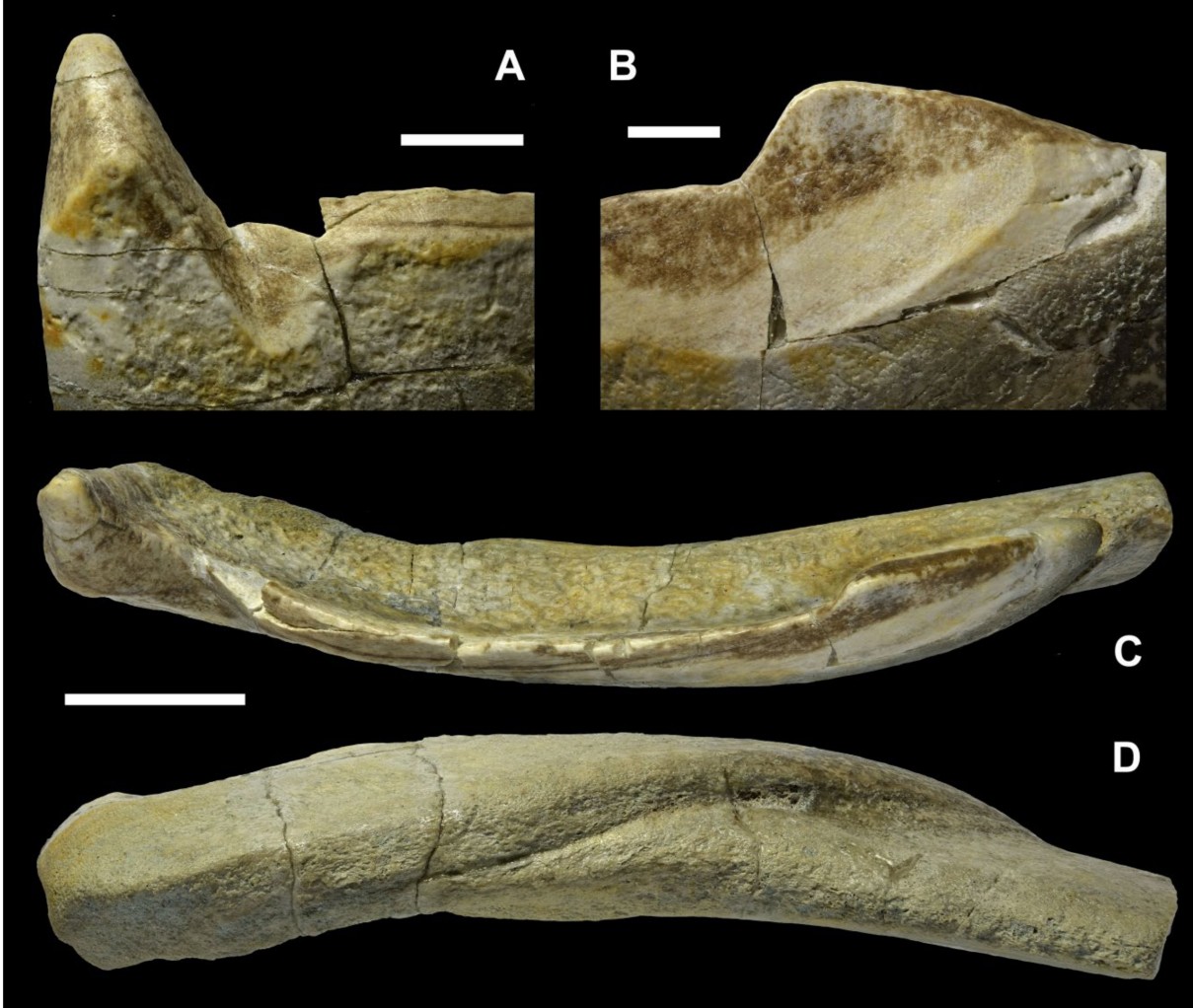

**Figure 15.** *Dunkleosteus tuderensis* Lebedev, sp. nov., holotype KMA 4155; (**A**) shows an enlargement of the anterior part of the occlusal margin including the symphyseal odontoid; (**B**) is an enlargement of the posterior corner of the occlusal margin, both in labial view. (**C,D**) present occlusal and ventral views of the holotype, respectively. Scale bar for (**A**) = 1 cm; for (**B**) = 5 mm, for (**C,D**) = 2 cm.

The ventral margin of the oral division is convex, and that of the preserved part of the adductor division is slightly concave. The antero-ventral flange and the Meckelian groove are mostly destroyed by post-sedimentary abrasion.

The lingual fossa is the most prominent structure on the lingual side; its maximum depth is approximately 3.5 times smaller than its length (Figure 14B). Most of its surface is covered by resorption bays (Howship's lacunae), being most expressed along the ventral margin of the fossa. Dorso-laterally, these pits are less expressed and are completely missing along the occlusal margin. The posterior corner of the occlusal margin forms a lingually overhanging swelling. The diamond-shaped contact area for the attachment of the mentomandibular cartilage, approximately four times longer than deep, extends along the antero-ventral margin. A smooth, slightly concave area of the anterior lingual process of the para-articular component is wedged between the lingual fossa and the attachment area for the mentomandibular cartilage.

## 5.2. Superficial Microstructure

Several areas within zones unaffected by wear or having almost no wear (Figure 15) were chosen to study the superficial structure of bone tissue. The walls of the incision (Figure 16A) produced by the anterior supragnathal odontoid are smoothly polished by the occluding odontoid of the anterior supragnathal, making it possible to see the layered bone texture. Closer to the labial surface, openings of the vascular canals are poorly organized. In the upper central part of the photo, natural brown staining reveals thin parallel bone layers. The surface of the occlusal margin presented in Figure 16B shows compact bone, including a network of thin vascular canals arranged longitudinally above the dark horizontal line of natural staining (formed along a fracture) and several large, looped canals below it. The surface of the posterior corner (Figure 16C) shows parallel lifetime wear traces most clearly; dark staining on top of the photo is secondary natural staining, mostly following bone cracks. An enlargement of the same area photographed under alcohol at larger magnification (Figure 16D) shows a layered structure with intermingled, occasionally anastomosing vascular canals.

The rest of the labial surface of the oral division is composed of highly porous bone. Ventrally from the symphyseal odontoid (Figure 16E), tiny vascular pores are mostly irregularly scattered on the bone surface, but sometimes become organized in short dorso-ventrally directed rows, or appear on the bottom of short grooves. Several larger vascular grooves of the same orientation enter the bone surface at an oblique angle (Figure 16F). Parallel to the postero-ventral margin of the oral division (Figure 16G), thin subparallel discontinuous layers of bone overlay each other at a small angle to the surface, so that their exposed edges form ridges. The bone layers are spaced, suggesting rapid growth. Dark spots are due to post-sedimentary natural staining. Rounded or ovoid pits with destroyed margins are formed by the pressure of sand particles. When visualized under alcohol (Figure 16H,H1), areas of white, unstained semi-translucent bone show outlines of osteocyte lacunae.

On the lingual surface of the oral division, the resorption bays (Howship's lacunae) cover the greatest part of the lingual fossa surface, being mostly expressed in its ventral and posterior parts. Sometimes, these pits are separated from each other, but in most cases they are fused, with their margins forming undulating unevenly spaced transverse rows (Figure 16I,J). All along the postero-ventral margin of the lingual fossa, there are elongated furrows marking the ventral limit of the resorption area (Figure 16K). Individual resorptive lacunae are ovoid or horseshoe shaped (Figure 17A–D). Visualization under alcohol, at higher magnification, enables the bone structure to be determined through the bottom of the lacunae, including individual bone layers, vascular canals and osteocyte spaces (bl, vc, oc, Figure 17C,D).

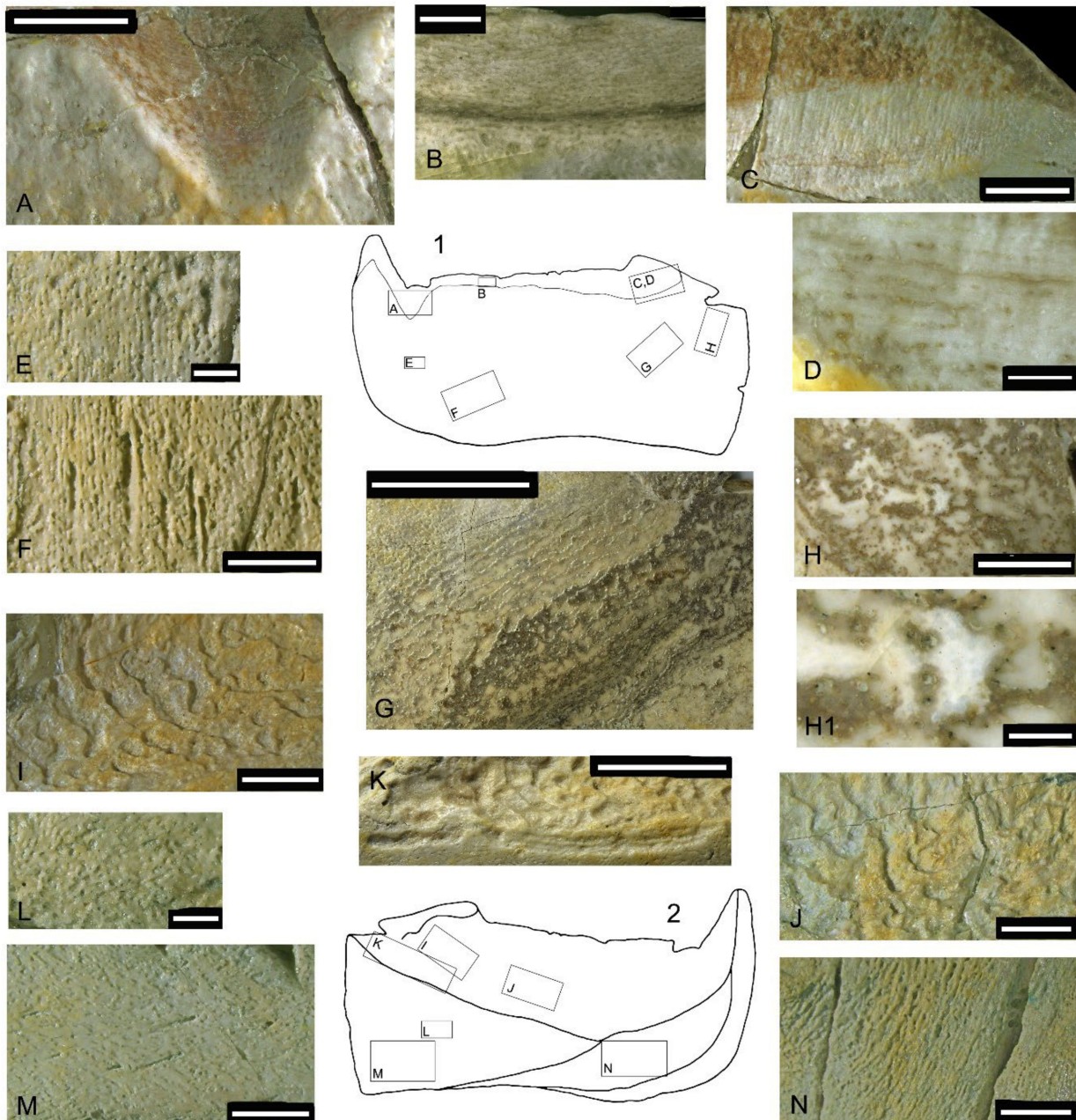

**Figure 16.** Details of superficial bone structure in *Dunkleosteus tuderensis* Lebedev, sp. nov., holotype KMA 4155, showing on the labial surface (**A**–**H1**) the walls of the incision produced by the anterior supragnathal odontoid (**A**); compact bone at the surface of the occlusal margin (**B**); the surface of the posterior corner of the occlusal margin (**C**) and its enlargement photographed under alcohol at larger magnification (**D**); an area ventrally from the symphyseal odontoid (**E**); bone surface showing grooves of vascular canals (**F**); a superficially lamellar structure of on the postero-ventral margin of the oral division (**G**,**H**); and the enlarged central part of the latter (**H1**). Those photographed under alcohol show white unstained semi-translucent bone areas presenting outlines of possibly osteocyte lacunae. On the lingual surface, three figures show various areas affected by resorption bays (**I**–**K**); the superficial structure of the anterior process of the para-articular component (**L**,**M**); and the roughly ornamented area of attachment of the mentomandibular cartilage (**N**). Scale bar for (**D**,**H1**) = 1 mm; for (**B**,**E**,**L**) = 2 mm; for (**A**,**C**,**F**,**H**–**J**,**M**,**N**) = 5 mm, for (**K**) = 10 mm and (**G**) =15 mm.

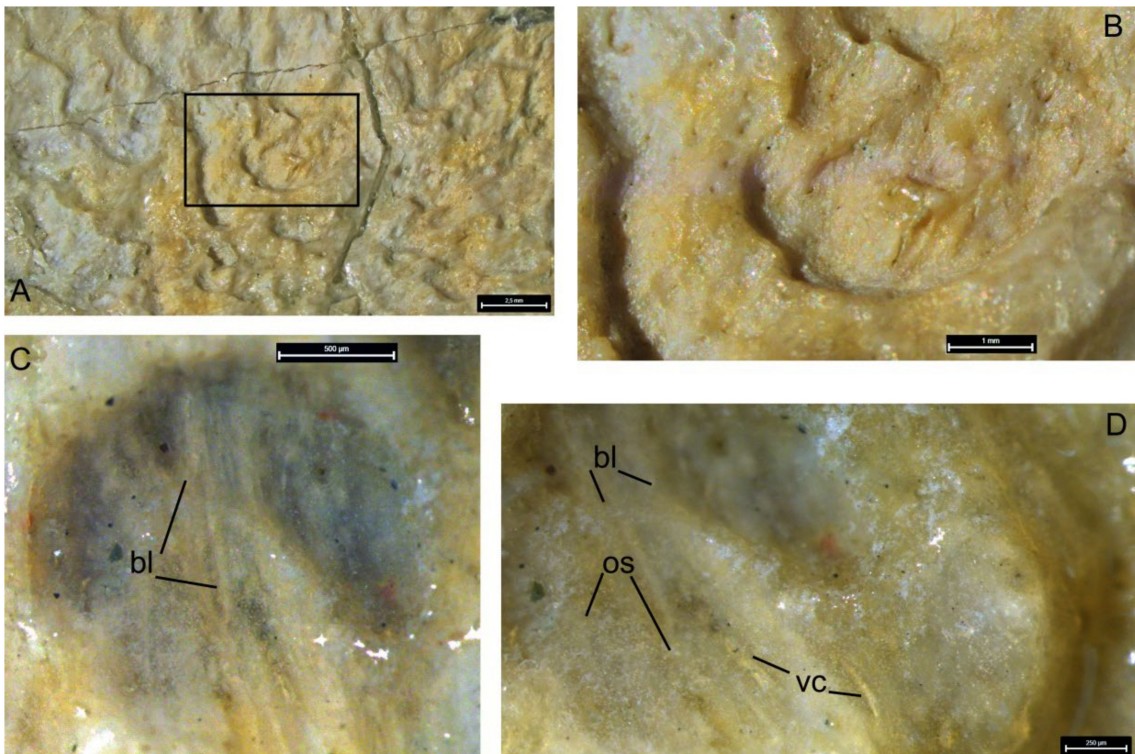

**Figure 17.** Details of superficial structure of resorption bays on the walls of the lingual fossa in *Dunkleosteus tuderensis* Lebedev, sp. nov., holotype KMA 4155 (**A**), showing an enlarged area (**B**), as well as the same under alcohol (**C**) and its enlargement (**D**). Abbreviations: bl, bone layers; oc, osteocyte spaces; vc, vascular canals.

The microstructure of the wedged area of the para-articular component forming the posterior part of the mouth cavity is reminiscent of that described above from the labial side in the area ventral to the symphyseal odontoid and parallel to the postero-ventral margin of the oral division. Here, numerous tiny vascular pores, forming sub-parallel oblique rows, sometimes lodge within minute grooves which possibly separate individual bone layers (Figure 16L). Small superficial grooves enter vascular canals running antero-dorsally within the bone mass, transversely to these bony rows (Figure 16M). The diamond-shaped contact area for the attachment of the mentomandibular cartilage bears rough, slightly oblique sub-vertical parallel bony crests separated by variously sized vascular foramina (Figure 16N).

*5.3. Internal Structure*

CT scanning of the holotype KMA 4155 revealed two distinct high-density constituents intermingled with bone tissue of low density (Figure 18A–J). These parts are generally comparable to the compact portions of the para-articular and oral components of the infragnathal described by Ørvig [46] in *Plourdosteus canadensis* and Rücklin et al. [52] in *Compagopiscis croucheri*. Unfortunately, low-density bodies cannot be reliably associated with high-density components because of the absence of a clear boundary in these tomographic slices. These less-dense regions may be interpreted as being composed of either partly resorbed tissue, or may instead represent preserved 'young', histologically immature bone which did not undergo compaction by deposition of lamellar bone in the lumina of vascular canals.

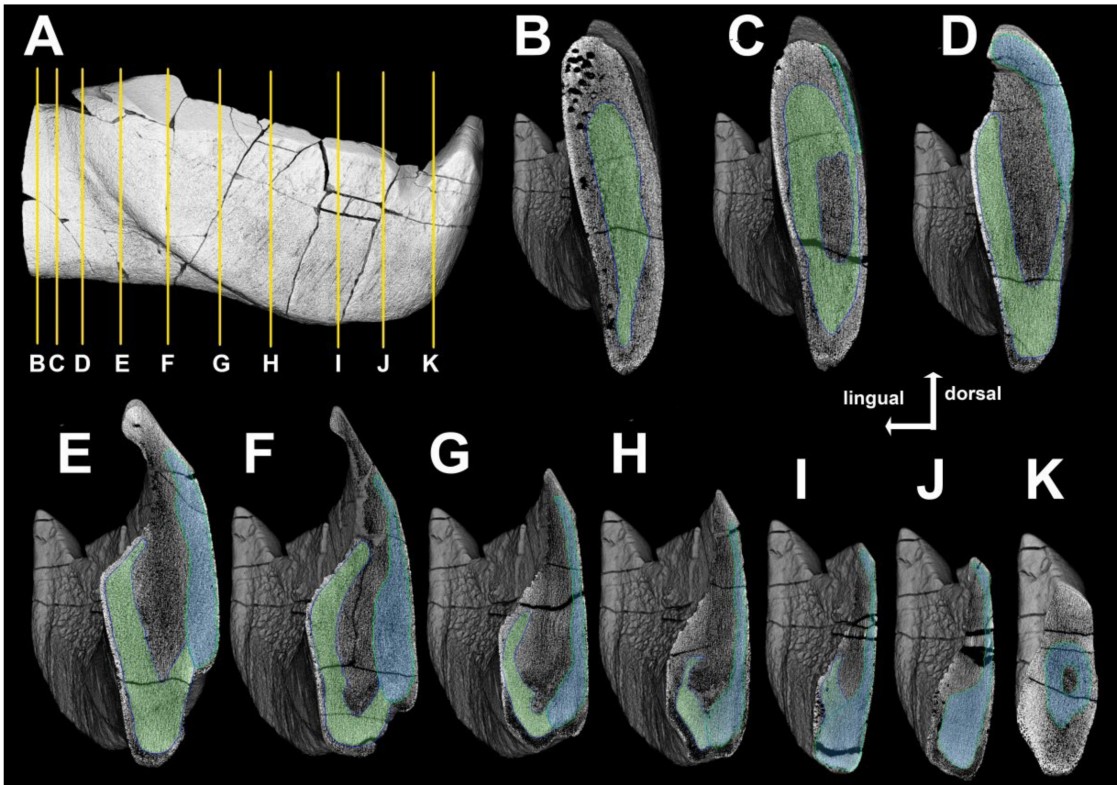

**Figure 18.** Virtual reconstruction of the infragnathal of *Dunkleosteus tuderensis* Lebedev, sp. nov., holotype KMA 4155, showing volumetric rendering of the infragnathal (**A**). Yellow lines indicate the position of sections, sections (**B–K**) show the inner structure of the infragnathal, green marks the compact part of the para-articular and the blue tone, the oral components. Low-density uncolored bodies cannot be reliably associated with high-density components.

Anteriorly, opposite to the symphyseal odontoid, the compact portions of these components seem to be fused, as no boundary between them is observable. Posterior to the fused region, the compact portions of the components closely adjoin each other ventrally in the midline, forming a trough filled with low-density bone (Figure 18D–I).

Anteriorly, the compact portion of the para-articular component consists of two parts (cpp, Figure 19A). The anterior part comprises a thin plate on the lingual side of the infragnathal ventral to the lingual fossa and posterior to the contact surface for the mentomandibular cartilage (acpp, Figure 19B,C,E). The posterior part forms an oblique furrow along the ventral margin. The lingual wall of this furrow follows the outline of the wedge-shaped superficially exposed process of the para-articular component, while the lateral wall roughly follows the posterior boundary of the oral component.

The internal compact part of the oral component (cpo) is a thin plate on the lateral face of the infragnathal following the outline of the oral division (Figure 18A,B,D,F). The ventral part of this plate gradually thickens anteriorly and terminates by a hollow cone supporting the symphyseal odontoid.

The bone density surrounding the compact parts is similar to that in the inner layer between them, except for the thin external layer forming the labial surface of the infragnathal. The bone tissue of this layer is most dense in the symphyseal odontoid, the lateral side of the dorsal part of the oral division and the anterior part of the para-articular component.

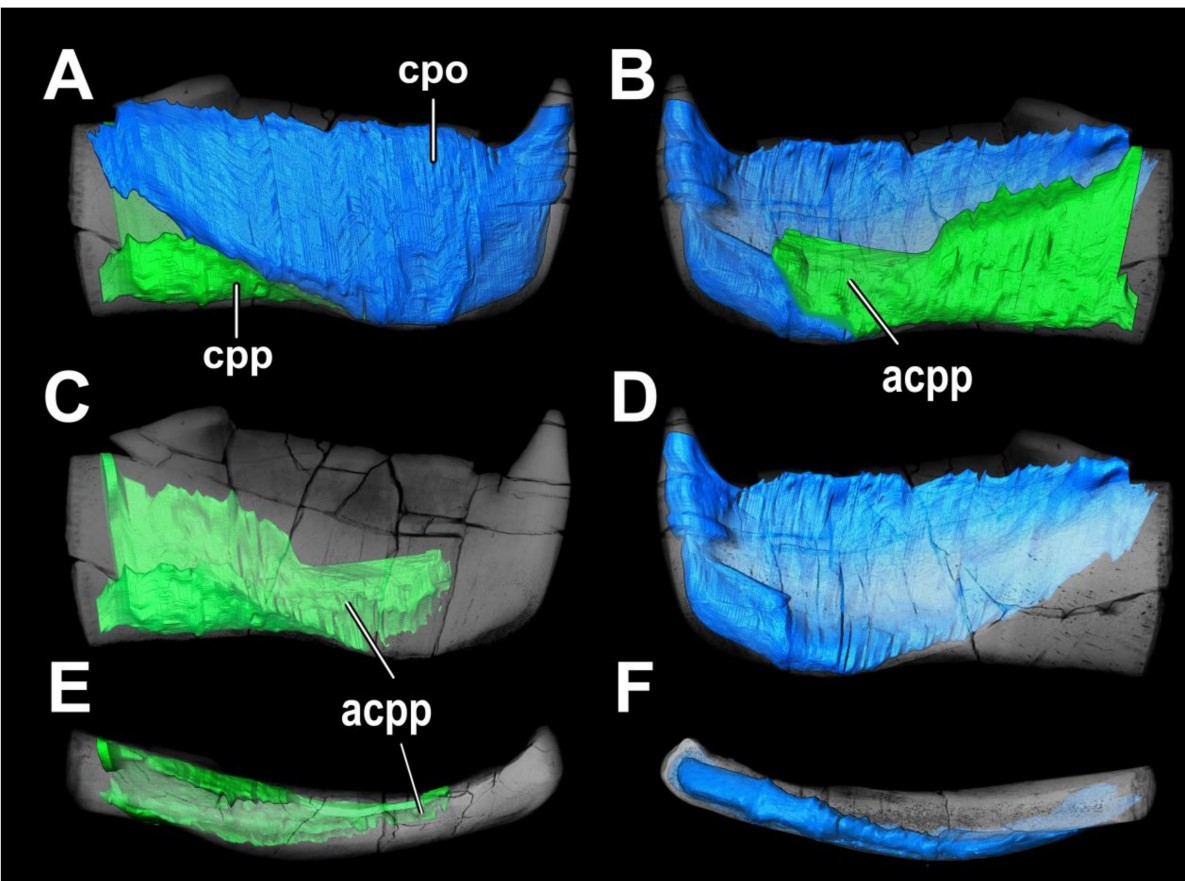

**Figure 19.** Virtual reconstruction of the infragnathal of *Dunkleosteus tuderensis* Lebedev, sp. nov., holotype KMA 4155, surface rendering of the compact inner parts of the oral and para-articular components: (**A**)—in labial view; (**B**)—in lingual view; (**C**)—in labial view, showing the compact part of the para-articular component; (**D**)—in lingual view, showing the compact part of the oral component; (**E**)—in dorsal view, showing the compact part of the para-articular component; (**F**)—in ventral view, showing the compact part of the oral component. Low density uncoloured bodies cannot be reliably associated with high-density components. Abbreviations: acpp, anterior part of the para-articular component; cpo, compact part of the oral component; cpp, compact part of the para-articular component.

*5.4. Vascular System*

Blood vessels supplying the bone and resorption areas run inside the infragnathal in regularly arranged rows of canals, both mesially and laterally to the compact parts of the para-articular and oral components (Figure 20; Supplementary Video S1). Canals in each row are parallel to each other, closely spaced and almost identical in thickness. Posterior to the oral component, the vascular canals are thicker and run almost parallel to the long axis of the jaw, before turning dorsally and opening at the posterior end of the lingual fossa. Only a few of these may be traced on the CT sections due to the high bone density in this region, and being plentiful posteriorly, this bunch of canals looks interrupted anteriorly. The oral component bears a series of thinner sub-parallel canals passing obliquely in a ventro-dorsal direction.

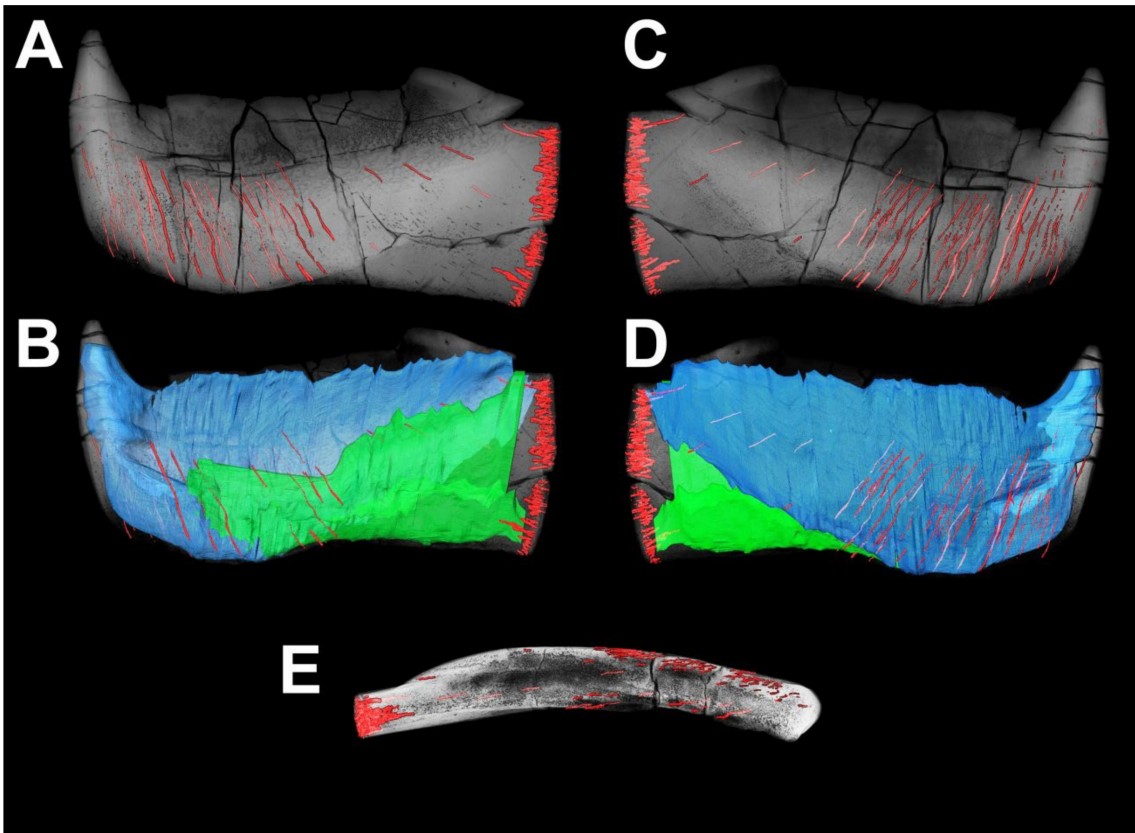

**Figure 20.** Virtual reconstruction of the vascular system (red) of infragnathal KMA 4155: (**A**,**B**)—in lingual view, (**A**)—with transparent volumetric rendering of the infragnathal, (**B**)—with surface rendering of compact parts of the components; (**C**,**D**)—in labial view, (**C**)—with transparent volumetric rendering of the infragnathal, (**D**)—with surface rendering of the compact parts of components; (**E**)—in ventral view with transparent volumetric rendering.

## 6. Discussion

### 6.1. Growth of Arthrodire Gnathal Elements

Little attention has been paid to the mode of growth in gnathal ossifications in arthrodires, with the exception of coccosteomorph arthrodires [52,83] and with rarer studies on pachyosteomorph arthrodires [12,84], all of which show a wide spectrum of variable morphologies.

Pioneering studies of gnathal growth were undertaken by Ørvig [46] who noticed superficially located subparallel grooves on the supra- and infragnathal elements of *Plourdosteus canadensis*. Interpretation of these grooves as growth lines suggested that the development of these elements started from the increase in bone trabeculae basally, with a successive transformation into more dense tissue and the simultaneous formation of teeth within the dental fields. Further basal apposition of new trabeculae, subsequently remodeled into compact bone, not only compensated for loss resulting from wear on the occlusal surfaces, but led to general enlargement of the element. Simultaneously, the osteosemidentine columns (OSC), composing the intermediate odontoids and worn apically, continued growing basally via transformation of the trabecular bone tissue. The twofold origin of the infragnathal was supported by evidence from thin sections showing a clear boundary between the two components fusing during ontogeny, which Ørvig [46] termed the 'axial' and the 'dental' (para-articular and oral, in this paper). It was hypothesized that fusion occurs by the lingual and labial margins of the oral component enclosing the axial component. Further growth included compensation for wear at the occlusal margin from the base of the oral component and its caudal extension. Skeletochronology indicated more active growth at the anterior portion of the growth zone [46]. This might result in gradual re-orientation, with relative 'lift' of the anterior part of the para-articular component with

respect to the symphyseal part of the oral component, and its incorporation into the occlusal margin with subsequent wear.

Hanke et al. [85] described the gross internal structure of an infragnathal, which these authors assigned to a new "dinichthyid" genus *Squamatognathus*. In the transverse section, they noted a distinct boundary between the two components differing in texture and orientation of the bone trabeculae ([85]: Figure 2E). The inner region consisted of less dense bone with more extensive vascular spaces; in the periphery, the bone is more dense, vascular spaces are smaller and the trabeculae are thicker and more numerous. Their observation corresponds to the state observed in *Plourdosteus canadensis* ([46]: p. 150, figure 24, 110) and suggests additional evidence for the presence of the oral and para-articular components in the infragnathal of *Squamatognathus*.

Rücklin et al. [52] reconstructed a virtual sclerochronological succession in the infragnathal of the coccosteomorph *Compagopiscis croucheri*, and found it was composed of two principal ossifications, the bony shaft of the jaw ('axial ossification') and a distal compound dental ossification. Their reconstruction of the developmental sequence demonstrated that growth of the oral component proceeded through the addition of new teeth starting from an original tooth position near the occlusal surface in three directions: symphyseally, medially and a marginal row posteriorly, the addition of which, at later stages, became associated with the growth of sheets of bony tissue extending ventrally around the bony shaft of the infragnathal, and in the largest infragnathal, partially around the Meckel's cartilage. These sheets are continuous and therefore are indicative of coordinated growth. Rücklin et al. [52] suggested the bony shaft is comparable to the inner jaw bones of the early osteichthyan fishes based on its position with respect to the Meckel's cartilage, overlying dental ossifications and the lateral attachment of adductor muscles.

Dennis-Bryan [12] mentioned that in a pachyosteomorph dunkleosteid *Eastmanosteus calliaspis*, the number of teeth in the symphyseal row grows with size increase. At earlier stages, the marginal tooth row extends along the whole length of the occlusal surface, while in the older individuals, only the posteriormost teeth remain while anteriorly a sharp edge replaces them. Carr [84] discussed the ontogeny of the infragnathal in the aspinothoracid *Heintzichthys gouldii* and commented that in the adult specimens, there is a single 'anterior cusp' (symphyseal odontoid) and the occlusal margin is smooth, while in the younger individuals, there is a row of teeth running along the occlusal margin. During life, these teeth become worn in the rostro-caudal direction, that is, most likely, new teeth had been formed posteriorly, reflecting the general growth direction.

The structure of the infragnathal in dunkleosteid pachyosteomorphs and some aspinothoracids differs from coccosteomorphs in features important from the point of view of growth pattern, although the general architecture remains the same. Common features include the location and mode of apposition of marginal and symphyseal tooth rows [20] and the interrelation of the oral and adductor division, making comparison possible. Similarly, the labio-basal deposition of highly vascularized bone in the infragnathals and basal (dorsal) in the supragnathals resulted in bone thickening. At the same time, in dunkleosteids, the lingual surfaces of the symphyseal odontoids of the infragnathals and the odontoids of the anterior supragnathals, as well as the buttresses formed by the OSC of the posterior supragnathals had not been affected by resorption but underwent strengthening by means of osteonal bone formation.

The OSC in the infragnathals and the posterior supragnathals in both groups are likely to be homologous. Notably, in those coccosteomorphs in which the medial tooth row is present on the infragnathal, such as *Compagopiscis* (=*Gogopiscis* in [20]: figure 7H), it occupies the same position as the OSC. Authors have suggested that the meeting point of the symphyseal, marginal and medial tooth rows is a primary growth center and the location of the first tooth primordium. Rücklin et al. [52] supported this idea. At the same time, the OSC occupies the same position as the medial tooth row and is an alternative to it. The difference between these dental structures is that the apex of the OSC is shifted backwards from the mesial corner of the infragnathal, but this may be due to its functional

wear that gradually obliterated the apex, disjointing it from the primordial position. In dunkleosteids and aspinothoracids, the OSC may be present or not, but the medial tooth row is always missing.

Unfortunately, the structure of the intermediate odontoid has never been studied in pachyosteomorphs, and our limited materials do not provide the opportunity to perform this histological study. We may only speculate here that its buttress is composed of the same tissue as the OSC within coccosteomorphs ([46]: figures 40–44).

In *Dunkleosteus tuderensis* sp. nov., there are no traces of any dentition, and the development of the OSC is minimal. In *Holdenius*, a short row of teeth runs along the dorsal margin of the non-occluding margin of the oral division, but these teeth face lingually. In *Gorgonichthys* and *Dinichthys herzeri*, short rows of marginal teeth are typically present only on the posterior slope of the posterior corner of the occlusal margin, as in some (not all) specimens of *Dunkleosteus terrelli*. A row of symphyseal teeth running along the symphyseal margin is also found in *Dinichthys herzeri*, but not in *Dunkleosteus terrelli*. Instead, as shown by Stetson ([19]: p. 27, pl. 5, figure 2) and Heintz ([10]: pl. 5, figures 12 and 13 and pl. 6, figures 14 and 15), in this species, the internal part of the symphyseal odontoid mostly consists of compact bone, but no dentine of any kind has been found. Our material from Dunkleosteidae indet. (Figure 6) supports their earlier observations.

For this reason, we suggest that in Dunkleosteidae, the formation of osteonal bone in the symphyseal odontoid, on the occlusal margin, and especially its posterior occlusal corner, undertook the function of harder dentinous tissues present in other jawed vertebrates, preserving the occlusal surface during feeding. We hypothesize that the symphyseal odontoid was formed as the ontogenetically oldest (thus more compact) part of the occlusal division of the jaw.

As seen in both species of *Dunkleosteus*, *D. terrelli* and *D. tuderensis* sp. nov., the result of bone remodeling with lingual resorption and lateral deposition results not only in the formation of an extensive and rather deep lingual fossa, but also in the development of postero-lateral thickening in the oral division of the infragnathal, thereby reinforcing the lateral wall of the oral cavity. The occlusal (shearing) facet is slanted labially, meaning newer bone layers (labially) are affected faster than the inner ones lingually. The same effect takes place in the posterior supragnathals, but in those, the occlusal facet faces lingually and harder osteonal bone forms the external, labial bone layers (Figure 11B,E,F). Reinforcement of the older lingual bone layers by the formation of new osteons may result in the formation of self-sharpening margins as a result of wear.

The odontoid apices showed evidence of wear due to pressure from contact with food items, becoming more rounded (i.e., blunter) with age. At the same time, both labial wear facets of the infragnathal symphyseal odontoids as well as internal facets of the anterior supragnathal odontoids display extensive wear surfaces, resulting in these bones sharpening the odontoid tips. Simultaneously, the anterior supragnathal odontoid, as well as the infragnathal symphyseal odontoid, slowly 'sawed' the bone in the oral–aboral direction at the base of the odontoid on the opposing jaw bone, forming a notch, and thereby causing an increase in the height of the upper and lower jaw odontoids. This process might have led to their relative elongation during life, supported by the symphyseal odontoids in the largest infragnathals of *Dunkleosteus terrelli* having proportionally larger bases.

### 6.2. Resorption Field

Resorption bays (Howship's lacunae) are cavities, pits or grooves formed by osteoclast cells acting on the surface of bone undergoing remodeling [86]. These structures on the gnathal bones of arthrodires were first described by Johanson and Smith [20]. In *Incisoscutum ritchiei*, *Goujetosteus* sp. and *Dunkleosteus* sp., these authors recorded the resorption and redeposition of tissues in active remodeling on the lingual side of the infragnathals. The described resorption bays vary in extent and depth. The resorption process is accompanied by deposition of new layers on the labial side of the jaw element. Remodeling inside the gnathal bone in *Dunkleosteus* occurs as new osteons fill in the

vascular canals and erosion bays ([20]: figure 19D,E), but, in contrast to *Antineosteus lehmani* ([20]: figure 19A–C), no evidence of the formation of pleromic dentine has been found in *Dunkleosteus* sp.

Apart from Johanson and Smith [20], no further studies of this peculiar process has been undertaken. Comparable, well-preserved resorption bays are present in *Dunkleosteus tuderensis* sp. nov. (Figures 16 and 17). Further examination of material from *D. terrelli*, made by one of the authors (R.K.E.) in the Cleveland Museum of Natural History, demonstrated these resorption fields on the walls of the lingual fossa are present in all examined specimens. In *Holdenius holdeni*, a small area, on the lingual side and ventral to the occlusal margin, also shows rows of resorption bays, but these are directed longitudinally ([69]: figure 1b), whereas in *Dunkleosteus*, they are directed transversally. Howship's lacunae were also identified in the aspinothoracidan *Heintzichthys* (CMNH 8096) and may be present in *Gorgonichthys* and *Bungartius* (though the condition in the latter two are unclear).

The function of the lingual fossa in the oral cavity is unclear. The question still remains as to what kind of soft tissue coated the fossa and how thick it was. It is clear this region was covered by some kind of perioral tissue. The presence of scale-like oral denticles on the lingual fossa of the early-middle Devonian *Squamatognathus* that are unfused to the underlying infragnathal [85] suggest this may be the ancestral condition, and later (late-middle–late Devonian) arthrodires replaced these structures with flat sheets of hardened perioral tissues, similar to how other groups of vertebrates evolutionarily replaced their teeth with keratinous beaks (e.g., dinosaurs or anomodont therapsids [87,88]) or fused tooth plates (e.g., chimaeroids or tetraodontiformes [89,90]). However, it is clear the tissue covering the sides of the gnathals in arthrodires could not have been either keratin or dentine. The texture of the bone has abundant resorption pits and generally does not resemble the texture in other placoderms proposed to have keratinous beaks [91]. Similarly, it is unlikely these regions were covered by fused sheets of dentine, as if this were the case, flattened sheets of mineralized tissue of unknown homology would be common finds alongside arthrodire fossils. Resolving this issue requires the discovery of more gnathal elements from geologically older (early to early-middle Devonian) arthrodires.

### 6.3. Microanatomy and Histology of the Gnathal Bones

The study of the histological structure of the gnathal bones performed on isolated materials from dunkleosteid pachyosteomorphs revealed the presence of several specific features in this group of arthrodires. Active bone remodeling proceeded in a specific way: despite modifications of the primary vascular canals (namely, their expansion with the formation of erosion bays and subsequent deposition of lamellar bone along its edges), the primary bone structure remained incompletely reworked and the primary original arrangement of vascular canals stayed intact (Figures 6–11). This contrasts with typical secondary bone remodeling (Haversian remodeling) in which several generations of secondary osteons are usually formed, resulting in a change in the primary bone structure (see [92], and references therein). Additionally, the primary location of vascular canals is not preserved during classical substitution by secondary bone.

An important histological feature of the skeleton of arthrodires is the presence of strongly vascularized bone [93]. This structure is distinctly different from the skeletal tissues of any living group of marine vertebrates: chondrichthyans have skeletons composed of cartilage, most teleosts (especially euteleosts) have some kind of acellular bone [94,95], and most marine tetrapods exhibit some kind of pachyostotic bone structure [96,97]. Instead, the histology of dunkleosteid bone more closely resembles that of terrestrial tetrapods, with distinct cortical and cancellous layers [1,93]; however, we note that crown-group cetaceans (but not "archaeocetes") also show this pattern [98]. This tissue includes a network of blood vessels forming a characteristic cancellous structure (Figures 6–11). This cancellous organization of the bone is potentially important in reducing the weight of the extensive and massive dermal skeleton. For example, despite the extensive dermal armor of *Dunkleosteus*

*terrelli*, the low density of these tissues suggest it only weighed ~7.5% of the animal's armor-free body mass ([1]: p. 30).

As previously noted by Carr et al. [45], the abrasion of the wear facets of gnathals accompanies the process of the osteonal bone development. The presence of this mechanism of strengthening of the occlusal bone surfaces of the gnathals was confirmed by our study (Figures 6–11). During the formation of osteonal bone, vascular canals became filled with lamellar bone tissue which strengthened the tissue and prevented infection from external pathogens. No invasion of pleromic dentine into the bone has been found, supporting the conclusions of Johanson and Smith [20].

Differences in the orientation of vascular canals between the oral and para-articular divisions of the infragnathal suggest the presence of at least two main arterial branches supplying this bone, anteriorly and posteriorly. These might correspond to the branches of the lateral hypobranchial artery, or the afferent mandibular artery positioned similarly in sharks, medially and ventrally to the Meckel's cartilage [99]. The posterior root, the branches of which might have entered the infragnathal from the inside of the adductor lamina, or from its lingual face, would have supplied most of the length of the para-articular component, facilitating resorption in the posterior part of the lingual fossa. The branches of the anterior root entered the infragnathal from below, opposite to the lingual process of the para-articular component, supplying the oral division, especially the resorption area at the anterior end of the lingual fossa.

## 7. Conclusions

The morphology of the gnathal apparatus of dunkleosteid arthrodire placoderms has been poorly studied, despite more than 150 years of research. Nomenclatorial problems with respect to these skeletal elements had never been addressed. For this reason, we propose standardizations based on morphological terms for several structures previously defined on functional grounds. Previous terms for structures of the occlusal margin have a very narrow odontological meaning, and using these terms creates morphological confusion. For this reason, we suggest adoption of the term "odontoid", used to refer to tooth-like projections on the jaw bones of some extant frogs, as a replacement term for the non-dental "cusps"/"fangs"/"tusks" of arthrodire gnathals. This term is based on the possible association of structures similar in external morphology and function but differing in histological structure and developmental origin.

KMA 4155 from the Lower-Middle Famennian of the Bilovo locality in the Tver Region of Russia, is assigned to a new species, *Dunkleosteus tuderensis* sp. nov. This specimen is an oral part of the infragnathal, characterized by a short symphyseal odontoid oriented perpendicular to an almost straight occlusal margin, a poorly developed intermediate odontoid with a hardly discernible buttress and non-differentiated anterior and posterior lingual fossae (of which the latter has no distinct shelf). These features allow KMA 4155 to be distinguished from other species of *Dunkleosteus* in which the infragnathal is known, including the type species, *D. terrelli*. Broader revisions of the many poorly characterized species referred to as *Dunkleosteus* are beyond the scope of this study.

CT scanning revealed two high-density constituents interpreted as compact portions of the para-articular and oral components, fusing during ontogeny. This supports earlier hypotheses on the dual nature of the infragnathal lower jaw element in arthrodires obtained from coccosteomorph gnathal elements and demonstrated in pachyosteomorph arthrodires as well.

Resorption bays (Howship's lacunae) are pits or grooves formed by osteoclast cells acting on the bone surface; in *Dunkleosteus*, those are located on the walls of the lingual fossa. These structures signal the presence of an active resorption zone, in parallel with zones of active bone growth, suggesting that growth of the infragnathal occurred via postero-lateral thickening of the oral division and elongation of this element during life.

CT scanning made possible, for the first time, the observation of numerous vascular canals running inside the specimen. The difference in orientation of their groups suggests the presence of at least two main arterial branches supplying this bone.

Histological analysis of the dunkleosteid arthrodire gnathals revealed extensive superficial resorption of bone within the lingual fossa of the infragnathal, accompanied by remodeling with retention of the primary arrangement of vascular canals in the inner part of the bone. The lifetime wear of occlusal facets accompanies the process of the osteonal bone development, but no pleromic dentine invasion has been found. The absence of pleromic dentine suggests the absence of migrating odontoblasts and dentine production in the gnathals of these fishes, possibly an autapomorphic feature distinguishing this group of arthrodires. The osteonal bone mechanically strengthened the bone tissue and prevented infection through the vasculature opened by wear. Reinforcement of the older bone layers at the lingual side of the occlusal margin via the formation of new osteons might have resulted in the formation of a self-sharpening effect of the blade during wear. Numerous remaining questions such as the details of osteonal bone formation, the origin and evolutionary development of the infragnathal, as well as the homology of its components should be resolved in future by studies of ontogenetic series of gnathal bones.

**Supplementary Materials:** The following supporting information can be downloaded at: https://www.mdpi.com/article/10.3390/d15050648/s1, Figure S1. Line drawings of KMA4155 showing areas of abrasion in lateral and medial views, upper and lower images respectively. Darker tone indicates more wear. Figure S2: Greyscale images of KMA4155, to present the specimen in higher resolution and contrast. (A), lateral, (B), dorsal (oral), (C), medial and (D), ventral views. Scale bar = 3 cm. Video S1: Video of KMA4155 showing internal vasculature of the infragnathal, generated from CT-scans.

**Author Contributions:** O.A.L. wrote a significant part of the text, including Introduction, Materials and Methods, Geographical and Geological setting, Systematic Paleontology, part of the Results and a significant part of the Discussion and Conclusions. R.K.E. wrote most of the section labelled General morphology and terminology in the Results, provided plentiful graphic materials on *Dunkleosteus terrelli*, and took an active part in the discussion of the paper from the beginning of these studies. P.P.S. performed histological studies and participated in the microanatomic analysis. V.V.K. prepared micro-CT reconstructions and wrote parts of the text regarding the internal structure of the infragnathal. V.V.L. kindly presented the infragnathal specimen KMA 4155 for our study. K.T. provided comparative morphological data and measurements for specimens from the Gogo Formation. Z.J. and M.M.S. contributing to writing and revising the manuscript. All authors have read and agreed to the published version of the manuscript.

**Funding:** This study was funded by the Russian Foundation for Basic Research (RFBR) and the Royal Society of London (RS), project number 21–54–10003.

**Institutional Review Board Statement:** Not applicable.

**Informed Consent Statement:** Not applicable.

**Data Availability Statement:** The holotype specimen of *Dunkleosteus tuderensis* Lebedev sp. nov. KMA 4155 is stored in the collection of the E. E. Shimkevich Andreapol Historical and Natural History Museum (Andreapol, Tver Region, Russia). Specimens from the Gornostayevka locality (Orel Region) are in the collection of the A.A. Borissiak Palaeontological Institute of the Russian Academy of Sciences (PIN RAS).

**Acknowledgments:** The authors acknowledge kind assistance from Roman A. Rakitov (PIN RAS), who performed the micro-CT scans. Macrophotography was carried out by Sergey V. Bagirov (PIN RAS). Special thanks to A. Pradel (MNHN, Paris, France) for sending us the photos of the infragnathal of *Dunkleosteus marsaisi*. We thank Roman Bapinaev (St. Petersburg State University) for making thin section photos. We thank C. Colleary, A. McGee and H. Majewski (CMNH), and J. Maisey and A. Gishlick (AMNH) for access to specimens in their care. We thank R. Carr, R. Drushel, J. Long, L.A. Rezanka and S. Simpson for helpful discussions regarding the functional morphology of dunkleosteid jaws, R. Boessenecker for providing photos of a cast of *Dunkleosteus marsaisi* at the CCNHM, and G. Young (MMMN) for providing photos of the holotype of *Squamatognathus steeprockensis*.

**Conflicts of Interest:** The authors declare no conflict of interest.

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
