# Peer review of "Structure, Growth and Histology of Gnathal Elements in Dunkleosteus (Arthrodira, Placodermi), with a Description of a New Species from the Famennian (Upper Devonian) of the Tver Region (North-Western Russia)"

_diversity, doi:10.3390/d15050648_

Round 1

Reviewer 1 Report

To the best of my knowledge and understanding, this paper is a much valuable addition to the field of arthrodire palaeobiology - one which I'd be happy to see published on Diversity after minor revisions (see the attached edited manuscript file for specific comments/corrections/suggestions). That said, I must stress the fact that I am not an expert on the materials and issues dealt with in Lebedev et al.'s manuscript, and as such, I strongly recommend an expert to revise is as well before proceeding to formal acceptance. 

Best regards, 

the reviewer

Author Response

Thank you to the reviewer for their careful review of our manuscript. We have tried to make all the requested changes, please see our responses on their annotated pdf. 

Reviewer 2 Report

Please find comments and suggestions in the attached file

Author Response

Thank you to the reviewer for their comments, we have tried to answer each comment, with our responses on the attached document.

Reviewer 3 Report

Dunkleosteus species are surprisingly under-researched in recent decades, with those published works mainly focus on the aspects such as the body-size and palaeoecology. This work on the contra, was done by experienced authors with proven accomplishment in the anatomical and histological investigation of early vertebrates. The investigation is sound and detailed, and is presented skillfully and well-illustrated. I am greatly impressed by the painstaking effort into this project, in fact the paper may be too long for “modern” standards, with contents sufficient to fill at least two papers, but that’s up to the authors and the editors of Diversity. All in all, I highly recommend this paper to be published as soon as possible.

There seem to be some formatting issues throughout the manuscript (many of them marked “(Error! Reference source not found”). I highlighted some, together with my point-to-point comments on some minor errors, in the sticky notes of the pdf. They are to be addressed before the acceptance.  While I enjoy following the thinking and inferring process of the authors, I find the discussion part can be more concise.

Author Response

Thank you to the reviewer for their comments. This reviewer suggested that we divide the paper and consider the section on the cranio-musculature of Dunkleosteus for a separate manuscript; we have done so. We have otherwise tried to address all his comments. 
